# Heterogeneity of hepatocyte dynamics restores liver architecture after chemical, physical or viral damage

Inmaculada Ruz-Maldonado[1,2,3,4,5], John T. Gonzalez [1,2,3], Hanming Zhang[1,2,3,5], Jonathan Sun[1,2,3,5], Alicia Bort[1,2,3,5], Inamul Kabir [6,7], Richard G. Kibbey[3,4], Yajaira Suárez [1,2,3,5], Daniel M. Greif [6,7] & Carlos Fernández-Hernando [1,2,3,5] ✉

Midlobular hepatocytes are proposed to be the most plastic hepatic cell, providing a reservoir for hepatocyte proliferation during homeostasis and regeneration. However, other mechanisms beyond hyperplasia have been little explored and the contribution of other hepatocyte subpopulations to regeneration has been controversial. Thus, re-examining hepatocyte dynamics during regeneration is critical for cell therapy and treatment of liver diseases. Using a mouse model of hepatocyte- and non-hepatocyte- multicolor lineage tracing, we demonstrate that midlobular hepatocytes also undergo hypertrophy in response to chemical, physical, and viral insults. Our study shows that this subpopulation also combats liver impairment after infection with coronavirus. Furthermore, we demonstrate that pericentral hepatocytes also expand in number and size during the repair process and Galectin-9-CD44 pathway may be critical for driving these processes. Notably, we also identified that transdifferentiation and cell fusion during regeneration after severe injury contribute to recover hepatic function.

The liver harbors vital homeostatic roles such as regulating the metabolism, detoxifying drugs and xenobiotics and supporting the immunological response against viral and bacterial infections[1,2]. Because of its anatomical location, the liver is continuously exposed to dietary components, toxic chemicals, and pathogenic agents that can potentially induce liver damage and failure. However, this organ can retain its integrity due to its extraordinary regenerative capacity[1,3]. Given this organ's unique features, it is not surprising that the location and cellular source of this promising hepatic reservoir with high plasticity have been a fundamental topic in liver biology for the past few decades. Numerous studies have shown not only metabolic

differences in hepatocytes along the portal-central axis known as liver zonation[4] but also heterogeneous populations of hepatocytes with differing proliferative rates[5-10]. Thus, the heterogeneous nature of hepatocytes within the liver lobule suggests a structured division of functions, including that of a regenerative niche by some authors. In this context, several studies showed that hepatocytes displayed stem-like qualities; *Wnt* signaling around the central vein[10] or a high telomerase activity distributed throughout the liver lobule[7] were involved in holding this plasticity.

On the other hand, other studies demonstrated that significant proliferative activity was originated from cholangiocytes, epithelial

[1]Vascular Biology and Therapeutics Program, Yale University School of Medicine, New Haven, CT 06520, USA. [2]Department of Comparative Medicine, Yale University School of Medicine, New Haven, CT 06520, USA. [3]Yale Center of Molecular and Systems Metabolism, Yale University School of Medicine, New Haven, CT 06520, USA. [4]Departments of Internal Medicine (Endocrinology) and Cellular & Molecular Physiology, Yale University, New Haven, CT, USA. [5]Department of Pathology, Yale University School of Medicine, New Haven, CT 06520, USA. [6]Yale Cardiovascular Research Center, Section of Cardiovascular Medicine, Department of Internal Medicine, Yale University School of Medicine, New Haven, CT 06511, USA. [7]Department of Genetics, Yale University School of Medicine, New Haven, CT 06511, USA. ✉e-mail: carlos.fernandez@yale.edu

cells that form the bile ducts[9]. Moreover, it was proposed the existence of a hybrid cell type that displays intermediate features between hepatocytes and cholangiocytes[11,12] and in models of severe liver injury, mechanisms of transdifferentiation between both cell types were described[9]. Despite this apparent lack of consensus about the cell type responsible for driving the proliferative stimulus and hepatocyte turnover in the damaged and healthy liver respectively, recent studies using lineage tracing models in mice point to subpopulations of mature hepatocytes to harbor these roles. Periportal (PP) hepatocytes have been suggested to undergo dominant self-expansion over peri-central (PC) hepatocytes post liver injury[6]. Contrary to these findings, it has been recently identified a major proliferative activity in mid-lobular (Mid) hepatocytes or hepatocytes located in zone 2 of the liver lobule compared to their PC or PP counterparts during liver home-ostasis and regeneration[5,8,13,14]. Interestingly, besides this regional pattern along the liver lobule, other authors have also uncovered differences in the proliferative rates of hepatocytes depending on their ploidy, with diploid hepatocytes having a growth advantage over polyploid hepatocytes for tissue replacement after chronic injury and homeostasis[5,15]. So, it seems that hepatocyte ploidy also plays a fundamental contribution to cell turnover and liver regeneration making this process multifactorial[5,15,16]. Altogether, these latest studies support that replication of pre-existing Mid hepatocytes is the primary mechanism through which the liver restores its mass after a lesion. Although the concept of predominantly Mid hepatocyte hyperplasia during liver homeostasis and regeneration is well established now, there are still some inconsistences about the involvement of other hepatocyte subpopulations to liver remodeling depending on the nature of the liver damage model that require additional examination. Moreover, hypertrophy and ploidy of hepatocytes or transdifferentiation during liver regeneration and repair have been little explored and nothing is known about liver remodeling and clonal expansion of hepatocytes in response to viral infection (coronavirus infection). Interestingly, the impact of cell fusion mechanisms during hepatic reconstitution, which may be crucial to restore liver function, has never been examined.

Here, we used a multicolor lineage tracing strategy to extensively quantify hepatocyte clonal expansion, hypertrophy and ploidy by tracking labeled hepatocytes developed during liver homeostasis and after chemical, physical, and biological injuries. Using histological techniques, we analyzed the repair response of hepatocytes and non-hepatocyte cells to recover the liver architecture after the different damages utilized in our study. Our observations confirmed that proliferation of Mid hepatocytes mainly maintains liver homeostasis and restores hepatic histology after chemical and physical injuries. Moreover, this subpopulation also combats liver mass loss after coronaviral infection with MHV-A59. Our work goes beyond other previous studies to demonstrate that in addition to hyperplasia of Mid hepatocytes, liver reconstitution is also accompanied by hypertrophy of Mid hepatocytes after chemical, physical, and viral damages. Of note, we identified the major clonal expansion and hypertrophy of hepatocytes following chronic exposure with CCl4, with the generation of large clones along the portal-central axis. In this injury model we also observed the major alteration in the hepatocyte ploidy, with diploidy prevailing over polyploidy. In addition, mechanisms of transdifferentiation and cell fusion were traced during liver regeneration after hepatic resection. We therefore mechanistically propose Galectin 9 (Gal-9)-CD44 signaling pathway to be critical for liver remodeling.

## Results

### Liver homeostasis is driven by proliferation of midlobular hepatocytes

We used three different mouse models to investigate clonal expansion of hepatocytes and their proliferative rates during liver regeneration and homeostasis. We employed multicolor lineage tracing Rosa26-Rainbow (*Rosa26[rbw]*) Cre-mediated recombination mice in our study. This model uses pairs of *LoxP* sequences to randomly recombine 3 of 4 fluorescent proteins (Fig. 1A)[17]. In the absence of Cre-recombinase-mediated recombination, CAG-EGFP is expressed constitutively resulting in each liver cell permanently expressing green color (Fig. 1A). We crossed these mice with TMX-inducible albumin Cre mice (*Alb-CreERT2*)[18] to generate our research model, *Alb-CreERT2 Rosa26[rbw]* mice, to specifically activate the Rainbow colors in cells expressing albumin, a gene differentially expressed in hepatocytes. Injecting the mice with 20 mg/ml TMX (3 consecutive days, 100 μl per day) resulted in hepatocyte-specific recombination throughout the liver lobule. With this approach, hepatocytes were permanently labeled and randomly recombined to express 1 of 3 colors, yellow (mOrange), red (mCherry) and/or light blue (mCerulean) (Fig. 1A). As Cre recombinase expression was restricted to hepatocytes, non-hepatocyte cells including cho-langiocytes, stellate cells, immune and endothelial cells remained labeled in green (CAG-EGFP) (Fig. 1A). Hepatocytes only expressed CAG-EGFP and not mOrange, mCherry, or mCerulean without TMX administration, confirming TMX strictly regulated Cre activity (Fig. 1A). 20 mg/ml TMX dose was determined after evaluation of labeling efficiency of both cell hepatocytes and non-hepatocytes in TMX dose-response tests (starting at 0.1 mg/ml up to 20 mg/ml per 1 day) (Supplementary Fig. 1A) and time-course tests (between 1 and 4 consecutive days) (Fig. 1A and Supplementary Fig. 1B). We histologically differentiated central veins from portal tracts due to the identification of CAG-EGFP stained bile ducts around portal veins (Fig. 1A) and CK19 IHC using consecutive slides (Fig. 1B). We confirmed central veins by IHC using consecutive slides for the marker glutamine synthetase (GS) that was expressed exclusively in the first 2–3 layers of hepatocytes sur-rounding the central veins (Fig. 1B).

One of the main features of the mouse liver is that 90% of hepa-tocytes are polyploid[15]. Numerous studies have demonstrated that the ploidy status of hepatocytes defines their proliferation rate and metabolic function[5,15,16]. Heterozygous multicolor reporter mice have been used to trace the fate of polyploid or diploid hepatocytes in vivo[5,19]. Given our *Alb-Cre Rainbow* mouse model was heterozygous, it allowed us to examine hepatocyte ploidy status. In a diploid hepato-cyte, one of the two pairs of chromosomes harbors a Rainbow allele, and the cell can express only one fluorophore (Fig. 1C). In contrast, polyploid hepatocytes can be labeled by co-expressing multiple colors (Fig. 1C).

We administrated TMX to heterozygous *Alb-Cre Rainbow* mice after the polyploidization period of hepatocytes[19], at 8 weeks of age. After a 2-week TMX washout period, hepatocytes were stochastically labeled with mOrange, mCherry, or/and mCerulean (Fig. 1A, quantifi-cation in 1D). Meanwhile, non-hepatocytes (labeled with CAG-GFP) took up a significantly reduced total area (Fig. 1D). Within the group of hepatocytes, 44% were labeled mCerulean, 26% mCherry and 30% mOrange (Fig. 1E). Next, we calculated the % of colored hepatocytes per zone within the liver lobule, identifying PP hepatocytes, Mid hepatocytes and PC hepatocytes (Fig. 1F). Midlobular hepatocytes expressing mOrange, mCherry or mCerulean were found significantly more numerous than PP or PC, with double the number of hepatocytes compared to the others (Fig. 1F). These results were in line with the number of cells per clone, which were notably more numerous in Mid areas formed mostly by individual hepatocytes followed by clones of 2 and 3 hepatocytes (Fig. 1G). Clones of 4 and 5 hepatocytes were less common and were mostly located in Mid and PC areas, reaching a relatively lower frequency than clone sizes formed by 1–3 hepatocytes (Fig. 1G, H).

Furthermore, we quantified basal levels of hepatocyte prolifera-tion during liver homeostasis, using the proliferative marker Ki67 through IHC on consecutive histological slides. Midlobular hepato-cytes showed a higher proliferative rate than PC or PP hepatocytes (Fig. 1I). Our observations agree with recent studies that found modest

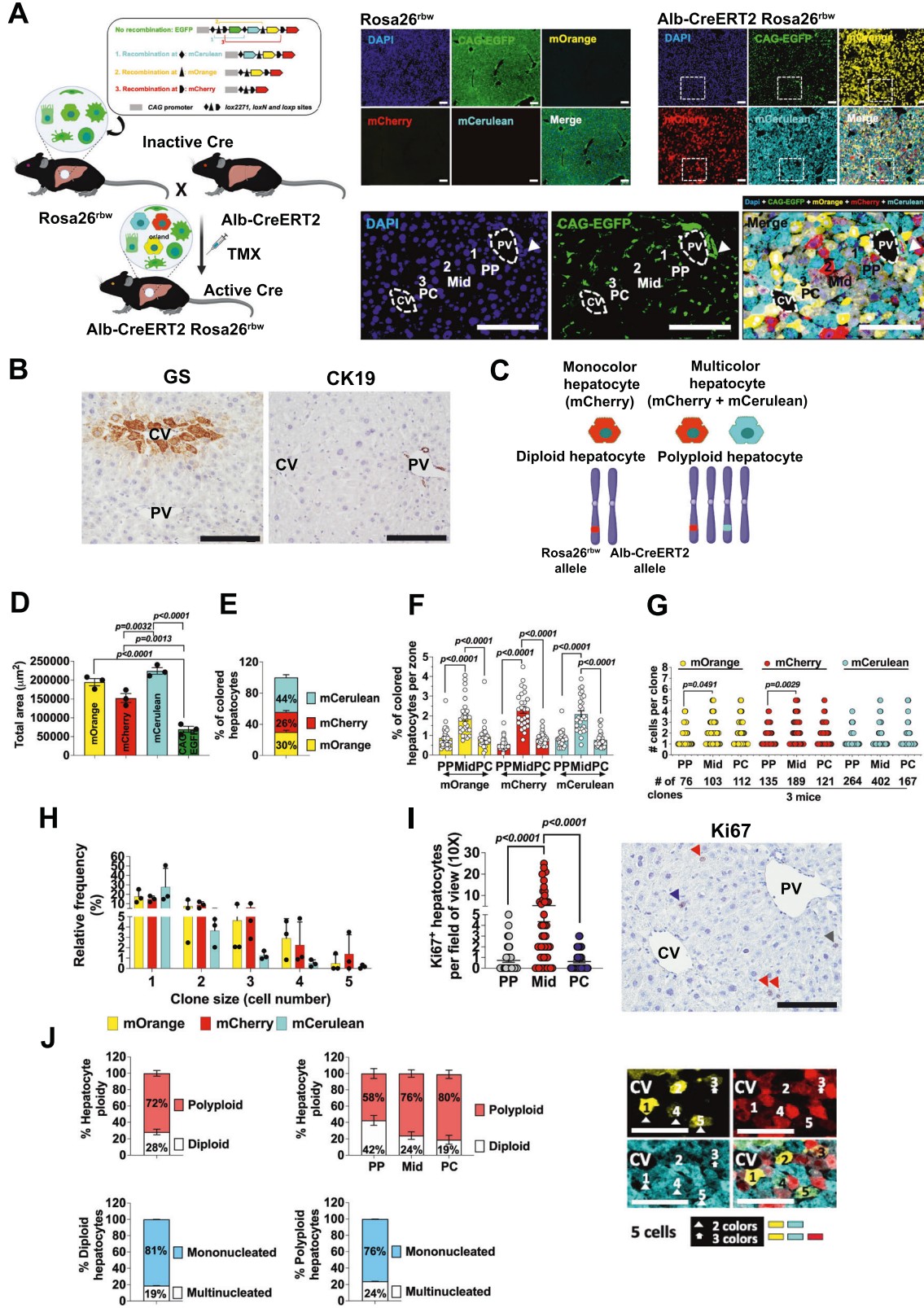

hepatocyte proliferation in all three zones, with more proliferation occurring in the Mid zone[5,8,13], and disagree with the prior concept of streaming of hepatocytes where new hepatocytes are generated in the PP areas and slowly expand toward the central vein[20].

As previous studies have found faster cell proliferation rates in diploid rather than polyploid hepatocytes[5,15], we wondered if an uneven distribution of ploidy across the liver lobule could account for

the increased proliferation in Mid hepatocytes. We, therefore, quantified the ploidy status of hepatocytes per zone of the liver lobule (Fig. 1J) to evaluate whether Mid hepatocytes exhibiting higher Ki67 staining were mostly diploid. We found that 72% of hepatocytes were polyploid and 28% diploid across the lobule in liver homeostasis of mice at 10 weeks of age (Fig. 1J). Within the liver lobule, polyploid hepatocytes were overrepresented in Mid and PC zones making up

**Fig. 1 | Hepatocyte renewal and ploidy during homeostasis in the healthy mouse liver. A** Schematic showing the 3 mouse models of our study; (Rosa26-Rainbow (*Rosa26^rbw*) Cre-mediated recombination mice, TMX-inducible albumin Cre mice (*Alb-CreERT2*) and *Alb-CreERT2 Rosa26^rbw* mice) (created with BioRender.com) and representative images of liver sections from 10-week-old male *Rosa26^rbw* and *Alb-CreERT2 Rosa26^rbw* mice (*n* = 4). The bottom panel shows a detail view of the liver lobule from a *Alb-CreERT2 Rosa26^rbw* male mouse, showing the 3 zones along the portal-central axis; zone 1 or periportal area (PP) that encircles the portal tracts identified by bile ducts (arrowhead); zone 3 or pericentral area (PC) that is located around central veins; and zone 2 or midlobular area (Mid) that is located in between. The Cre-activated fluorophores (CAG-EGFP, mCerulean, mOrange and mCherry) plus DAPI staining are shown in the images. CV central vein, PV portal vein. TMX tamoxifen. Scale bars, 100 μm. **B** Representative Glutamine Synthetase (GS) and Cytokeratin 19 (CK19) IHC analysis in liver sections from a 10-week-old *Alb-CreERT2 Rosa26^rbw* mouse (*n* = 4). Scale bar, 100 μm. **C** Schematic of the heterozygous multicolor *Alb-CreERT2 Rosa26^rbw* mice, that allow us to trace the fate of polyploid or diploid hepatocytes in vivo. Created with BioRender.com. **D** Total area in μm² per fluorophore and per photo or field of view (10×) of liver sections from *Alb-CreERT2 Rosa26^rbw* mice. 6–7 photos per mouse; *n* = 3 mice. Data are presented as mean values +/− SEM. One-way ANOVA, Tukey's multiple comparisons post-test. **E** Colored hepatocytes in % per Rainbow fluorophore and per photo or field of view (10×). Data are presented as mean values +/− SEM. 6–7 photos per mouse; *n* = 3 mice. **F** Colored hepatocytes in % per Rainbow fluorophore and per area of the liver lobule (PP, Mid and PC). 27 liver lobules analyzed from 3 mice. Data are presented as mean values +/− SEM. One-way ANOVA, Tukey's multiple comparisons post-test. **G** Quantification of number of cells per clone (y-axis) and number of clones (x-axis) per area of the liver lobule and per Rainbow fluorophore. Data are presented as mean values. 32 liver lobule areas analyzed from 3 mice (7–8 photos per mouse). One-way ANOVA, Tukey's multiple comparisons post-test. **H** Relative frequency in % of each clone size per area of the liver lobule and per Rainbow fluorophore. Data are presented as mean values +/− SEM. 32 liver lobule areas analyzed from 3 mice (7–8 photos per mouse). **I** Number of total Ki67 positive hepatocytes per area within the liver lobule and per photo or field of view (10×). 48 photos analyzed from 3 mice. Data are presented as mean values +/− SEM. One-way ANOVA, Tukey's multiple comparisons post-test. Ki67 IHC showing a liver lobule. Gray, red and blue arrows show PP, Mid and PC hepatocytes positive for Ki67. Scale bar, 100 μm. **J** Hepatocyte ploidy in the liver lobule of 10-week *Alb-CreERT2 Rosa26^rbw* mice. % of total polyploid and diploid hepatocytes within the liver lobule (top left graph) and per area of the liver lobule (top right graph). % of diploid hepatocytes (hepatocytes that express only 1 fluorophore) being mononucleated or multinucleated (bottom left graph). % of polyploid hepatocytes (hepatocytes that express more than 1 fluorophore) being mononucleated or multinuclated (bottom right graph). Data are presented as mean values +/− SEM. Right: Immunofluorescence showing 4 of 5 individual PC hepatocytes expressing more than 1 color. 5 photos from 3 mice were analyzed. Scale bar, 100 μm. Source data are provided as a Source Data file.

76% and 80% respectively compared to PP areas, where 58% of the hepatocytes were polyploid. (Fig. 1J). Even though 81% of diploid hepatocytes were mononucleated, 76% of polyploid hepatocytes were mononucleated as well (Fig. 1J), so nuclear ploidy did not match with the hepatocyte ploidy status in our model. We classified as polyploid hepatocytes those which expressed more than one fluorophore.

## Periportal and midlobular hepatocytes repopulate the liver after CCl4-induced acute injury

We next investigated the response of hepatocytes to acute liver injury with CCl4 and their implication in the remodeling of the whole organ mass recovery. CCl4 is a hepatotoxic compound that causes cellular damage to PC hepatocytes as they express the cytochrome P450 enzyme Cyp2e1, which is required to metabolize CCl4 into reactive free radicals[21]. As shown in Fig. 2A, a single dose of CCl4 was administrated in *Alb-CreERT2 Rosa26^rbw* mice and liver histology was examined at 2 and 12 days after the injection. We found severe liver injury at day 2 after CCl4 treatment by examining liver sections using H&E staining, hepatocyte (mOrange+, mCherry+ or/and mCerulean+ cells) and non-hepatocyte (CAG-EGFP+ cells) immunofluorescence (IF) analysis and GS IHC (Fig. 2B, C). Approximately 2/3 of the hepatocyte mass was depleted, with almost no colored-hepatocytes detected at the PC area and consecutive Mid regions compared to control livers (Fig. 2C, D). H&E staining showed necrotic/apoptotic hepatocytes in PC and Mid regions (Fig. 2B), which correlated with a significant accumulation of TUNEL positive cells (Fig. 2G). GS expression in PC hepatocytes was also reduced at day 2 of CCl4 compared to livers from untreated mice (Fig. 2B). Despite this extensive damage caused by CCl4, half of the Mid hepatocytes and the entire PP hepatocyte population remained morphologically normal (Fig. 2B–C). Notably, we observed fewer individual hepatocytes in these areas comparing to control mice (Fig. 2D and Fig. 1G), predominating clones formed by 2–4 hepatocytes (Fig. 2D) and numerous 5 hepatocyte clones (Fig. 2D). In line with these results, the percentage of the relative frequency of clones formed by 1 hepatocyte was reduced compared to control mice while that of 2–4 hepatocytes increased at day 2 after CCl4 (Figs. 2E and 1H). Clones formed by 5 or more hepatocytes were significantly increased over control mice, with a relative frequency of 5% (Fig. 2E) compared to <1% in livers from control mice (Fig. 1H). Twelve days after the initial CCl4 injection, the liver was completely recovered showing a normal hepatocyte morphology by H&E and IF analysis (Fig. 2B, C). GS positive hepatocytes were nearly restored (Fig. 2B) and dead hepatocytes

measured through TUNEL staining were barely detected (Fig. 2G). Clones at PC areas were identified de novo mainly formed by ≥2–4 hepatocytes (Fig. 2D), with a relative frequency of 6–7% (Fig. 2E). Larger clones of 5 hepatocytes were also found although their relative frequency was less abundant than smaller clones and accounted for 4% of the population (Fig. 2D, E). Interestingly, we detected clones formed by 6 hepatocytes only at 12 days after CCl4 treatment. They were concentrated in Mid and PC areas while PP zones mainly were composed of 3–4 hepatocyte clones (Fig. 2D, E). The differences observed in colored-hepatocyte dynamics within the liver lobule at days 2 and 12 after CCl4 suggested that hepatocytes of the 3 zones experienced different proliferation rates to recover lobule integrity. So, we next investigated the location of hepatocytes entering the cell cycle by IHC staining of the proliferative marker Ki67 in consecutive liver samples. Immunostaining for Ki67 showed significantly higher Ki67+ hepatocyte % at day 2 after CCl4-induced damage compared to livers from control mice or mice at day 12 after CCl4 injection (Fig. 2G, I). Notably, we only found proliferating Ki67 hepatocytes in PP and Mid zones with no Ki67+ staining for hepatocytes around central veins (Fig. 2G, I). PP hepatocytes showed a higher proliferative rate than Mid hepatocytes at day 2 after CCl4 injection although it failed to reach the statistical significance (Fig. 2G, I). However, 24 h later, on day 3 after CCl4 treatment, individual PC hepatocytes and some PC hepatocyte clones were generated, as shown in Supplementary Fig. 7A. Three days later, at 6 days after the initial injection with CCl4, larger PC and Mid hepatocyte clones were observed (Supplementary Fig. 7B) and Mid hepatocytes reached their highest proliferative activity at this time point (Supplementary Fig. 7B, C). The number of proliferating Ki67 hepatocytes was drastically reduced in the Mid area after 12 days of CCl4 treatment, with a mean of 0.4 ± 0.16 Ki67+ hepatocytes per field of view while we did not practically find any proliferating Ki67 hepatocyte in PC or PP zones (Fig. 2G, I). These data demonstrate that undamaged Mid hepatocytes repopulate the liver after CCl4-induced acute injury via proliferation giving rise to new hepatocytes which will displace adjacent pre-existing hepatocytes to central vein areas. Displaced hepatocytes acquired their zonated metabolic function as illustrated by GS re-expression on day 12 after CCl4 treatment (Fig. 2B).

The proportion of non-hepatocytes, tracked by CAG-EGFP staining was significantly increased at day 2 after CCl4 treatment compared to control mice and mice sacrificed after the recovery period at day 12 after the injury (Fig. 2F). Since CCl4 toxicity causes macrophages and monocytes infiltration which facilitate the removal of cellular debris

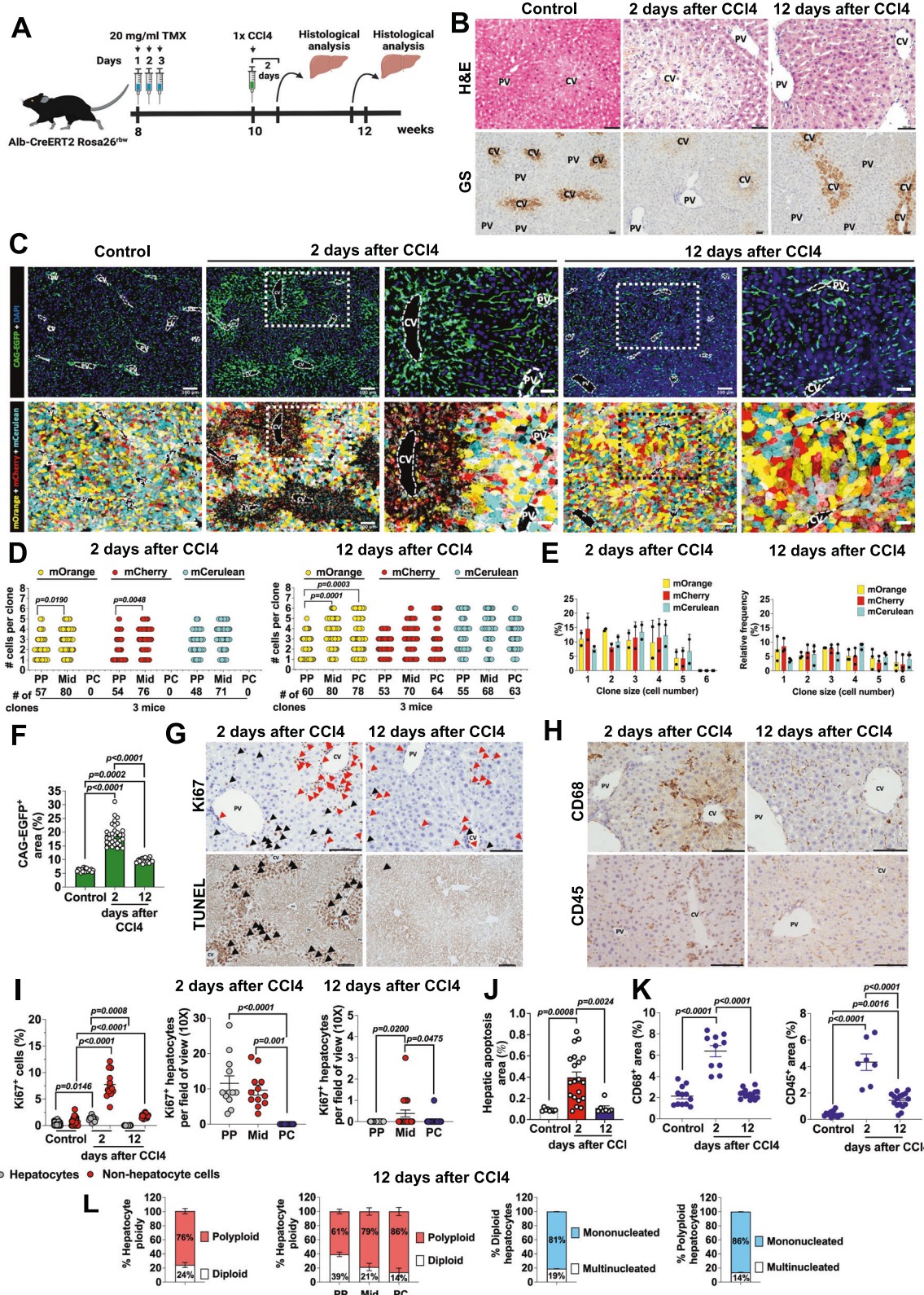

generated by necrosis and apoptosis of hepatocytes, we next examined leukocyte and macrophage infiltration during liver repair. To this end, we performed IHC analysis of CD68+ (macrophage marker) and CD45+ (leukocyte marker) cells at early (2 days) and late stage (12 days) of liver regeneration after CCl4 -induced damage. We observed a marked accumulation of CD68+ macrophages and CD45+ cells around central veins 2 days after CCl4 injection (Fig. 2H, K). Ten days later,

macrophage area was markedly reduced although the influx of CD45 immune cells was still significantly elevated with respect to control mice (Fig. 2H, K). Finally, we quantified the ploidy status of hepatocytes in each zone of the liver lobule after 12 days of CCl4 treatment to evaluate whether hepatocytes generated de novo in PC and Mid areas during the recovery phase differed in ploidy from control mice (Fig. 2L, Fig. 1J). We found that 76% of hepatocytes are polyploid and

**Fig. 2 | Periportal and midlobular hepatocytes repopulate the liver after CCl4-induced acute injury.** **A** Schematic showing the experimental design for CCl4-induced liver injury. Created with BioRender.com. **B** Representative H&E and GS staining of liver sections from untreated (control) and CCl4 treated mice ($n = 4$). Scale bars, 100 μm. **C** Representative immunofluorescence images of the liver lobule architecture displaying pericentral damage at 2 days post-CCl4 and liver repair at 12 days post-CCl4 ($n = 4$). Scale bars, 100 μm. **D** Quantification of number of cells per clone (y-axis) and number of clones (x-axis) per area of the liver lobule and per Rainbow fluorophore. Data are presented as mean values. 11–13 liver lobule areas were analyzed per condition (3–7 photos per mouse); 3 mice per group. One-way ANOVA, Tukey's multiple comparisons post-test. **E** Relative frequency of each clone size per area of the liver lobule and per Rainbow fluorophore. Data are presented as mean values +/− SEM. 11–13 liver lobule areas were analyzed per condition (3–7 photos per mouse); 3 mice per condition. **F** Quantification of the % positive area per field of view (10×) of CAG-EGFP in liver sections from untreated and CCl4 treated mice. Data are presented as mean values +/− SEM. 8 photos per mouse; 3 mice per group. One-way ANOVA, Tukey's multiple comparisons post-test. **G** Ki67 IHC and TUNEL assay in mouse liver sections at 2 and 12 days after CCl4. Ki67: red arrows indicate non-hepatocyte cells positive for Ki67 while black arrows show proliferating Ki67 hepatocytes. TUNEL: black arrows indicate dead hepatocytes. Scale bars, 100 μm. **H** CD68 and CD45 IHC of liver sections from mice at 2 and 12 days after CCl4. Scale bars, 100 μm. **I** Quantification of the Ki67 staining. Left graph: Total Ki67 positive cells (hepatocytes and non-hepatocyte cells) in % per field of view (10×) of liver sections from mice at 2 and 12 days after CCl4 treatment. 27, 12 and 23 photos from 3 mice for the untreated, 2 days post-CCl4 and 12 days post-CCl4 groups, respectively. Middle graph: proliferating Ki67 hepatocytes per field of view (10×) in each zone of the liver lobule (PP, Mid and PC) after 2 days of CCl4 exposure. 12 photos from 3 mice. Right graph: proliferating Ki67 hepatocytes per field of view (10×) in each zone of the liver lobule after 12 days of CCl4. 23 photos from 3 mice. Data in all panels are presented as mean values +/− SEM. One-way ANOVA, Tukey's multiple comparisons post-test. **J** % Apoptosis+ area from the images shown in section G. Data are presented as mean values +/− SEM. 7–8 photos per mouse; $n = 3$ mice per group. One-way ANOVA, Tukey's multiple comparisons post-test. **K** Quantification of the CD68 and CD45 IHC. CD68 and CD45 area in % per field of view (10×) in liver sections from untreated control and CCl4 treated mice. CD68: 10–18 photos were analyzed per mouse; $n = 3$ mice per group. Data are presented as mean values +/− SEM. CD:45: 7–18 photos per mouse; $n = 3$ mice per group. Data are presented as mean values +/− SEM. One-way ANOVA, Tukey's multiple comparisons post-test. **L** Analysis of hepatocyte ploidy in the liver lobule from mice after 12 days recovery post-CCl4 administration. % of total polyploid and diploid hepatocytes within the liver lobule (left graph) and per area of the liver lobule (middle left graph). % of diploid hepatocytes (hepatocytes that express only 1 fluorophore) being mononucleated or multinucleated (middle right graph). % of polyploid hepatocytes (hepatocytes that express more than 1 fluorophore) being mononucleated or multinucleated (right graph). 5 photos per mouse; $n = 3$ mice. Data are presented as mean values +/− SEM. Source data are provided as a Source Data file.

24% diploid within the liver lobule (Fig. 2L), similar values to those found in liver homeostasis (Fig. 1J). Like in control mice, Mid and PC zones resulted in having a majority of polyploid hepatocytes, content with a 79% and 86% respectively compared to PP areas, where 61% of the hepatocytes were polyploid. However, PP was mostly diploid (39%), followed by Mid (21%) and PC (14%) (Fig. 2L). Although 81% of diploid hepatocytes were mononucleated, 86% of polyploid hepatocytes turned out to also be mononucleated (Fig. 2L). So, we confirm with this model of CCl4 damage the previous findings in healthy livers: that nuclear ploidy does not match hepatocyte ploidy (Fig. 1J).

## Midlobular hepatocytes remodel the liver architecture after CCl4-induced chronic injury

We next analyzed liver lobule changes in response to chronic CCl4 lesion over time (Fig. 3). Ten-week-old *Alb-CreERT2 Rosa26^rbw* mice received 12 injections of CCl4 or corn oil every 2 days for 4 weeks and then they were sacrificed at weeks 14, 16 and 20 of age (Fig. 3A). Mice that received CCl4 treatment experienced a gradual elevation in their body weight over the course of additional injections with CCl4 (Supplementary Fig. 3A). After the sacrifice, we also observed a positive trend in the spleen weight in CCl4-treated animals, compared to control mice, immediately after CCl4 treatment cessation, at week 14, but they did not reach statistical significance (Supplementary Fig. 3B). Nevertheless, liver weight was remarkably increased in CCl4-treated mice at week 14, followed by week 16, whereas, at week 20, liver weight from control and CCl4 mice had similar values (Supplementary Fig. 3B). As it is shown by H&E and Sirius red staining, additional doses of CCl4 caused chronic injury, that was confirmed by significant pericentral fibrosis in mice sacrificed at week 14 (Fig. 3B, E). Fibrosis area decreased throughout the following two time points, with week 16 showing significant reduction and week 20 nearing the values of livers from control mice (Fig. 3B, E and Supplementary Fig. 3C). These data were consistent with the non-hepatocyte CAG-EGFP staining, with a positive area significantly increased at week 14 but markedly reduced over time during the recovery periods at weeks 16 and 20 (Fig. 3C, G). Our IHC staining also revealed that GS hepatic expression also was changed upon CCl4 exposure since PC hepatocytes reduced the expression of this enzyme at week 14 but this was entirely recovered during the repair period at week 20 (Fig. 3B), with a similar expression pattern to that found in livers from control mice (Fig. 1B). However, the expression of the cholangiocyte marker CK19 showed no alteration at week 14 after CCl4 repeated exposure and at weeks 16 and 20 during the recovery periods (Supplementary Fig. 3D). Chronic CCl4 damage promoted the development of large hepatocyte clones throughout the liver lobule compared to control mice (Fig. 3C, H and Supplementary Fig. 3E, F). After 12 doses of CCl4, we found large clones formed by 8–12 hepatocytes along the 3 zones of the liver lobule in mice sacrificed at week 14 (Fig. 3C, H). At week 20, larger-sized clones formed by 24–16 hepatocytes covered the entire distance from the central to portal veins (Fig. 3C, D, H). We also observed large clones that were extended from the portal triad areas to each other and some isolated medium-sized clones in the Mid zones (Fig. 3D). Clones formed by more than 10 hepatocytes were predominated compared to control mice (Supplementary Fig. 3F), with a relative frequency of 5% (Fig. 3F) versus 0.2% (Supplementary Fig. 3F). Analysis of hepatocyte area at week 20 also showed that large hepatocytes were found within the clones in all 3 different areas of the liver lobule, suggesting that they also exhibit mechanisms of hypertrophy in addition to proliferative events (Fig. 3I, Supplementary Fig. 9). In terms of hepatocyte ploidy status, 66% of the hepatocyte population generated were diploid while 34% of hepatocytes were polyploid (Fig. 3P). These data are significantly different from the hepatocyte ploidy found during liver homeostasis in the healthy liver (Fig. 1J), or liver regeneration in mice treated with only one dose of CCl4 (Fig. 2L). These displayed around 70% polyploid and 30% diploid hepatocytes (Fig. 2L). At week 20, we also found differences in hepatocyte ploidy within the 3 zones of the liver lobule, with more numerous diploid hepatocytes in PP (84%) and Mid areas (71%) and fewer polyploid hepatocytes in the PC area (39%) (Fig. 3P). Although 93% of diploid hepatocytes were mononucleated, 90% of polyploid hepatocytes were mononucleated as well (Supplementary Fig. 3H), similar to values observed in livers from control mice (Fig. 1J) and those that received acute CCl4 damage (Fig. 2L).

We next examined the proliferation of hepatocytes after extended CCl4 treatment. Immunostaining for Ki67 showed a significantly higher percentage of proliferating Ki67+ hepatocytes at week 14 after additional doses of CCl4 compared to control mice (Fig. 3J, K). Mainly, we found significantly more Ki67+ hepatocytes per field of view in the Mid and PC areas than in the PP region at this time point (Fig. 3L). However, we did not identify any proliferating Ki67+ hepatocytes during the recovery times at weeks 16 and 20 or after 2- and 6-weeks post damage respectively (Fig. 3K and Supplementary Fig. 3G). We also quantified liver apoptosis levels through TUNEL staining after chronic

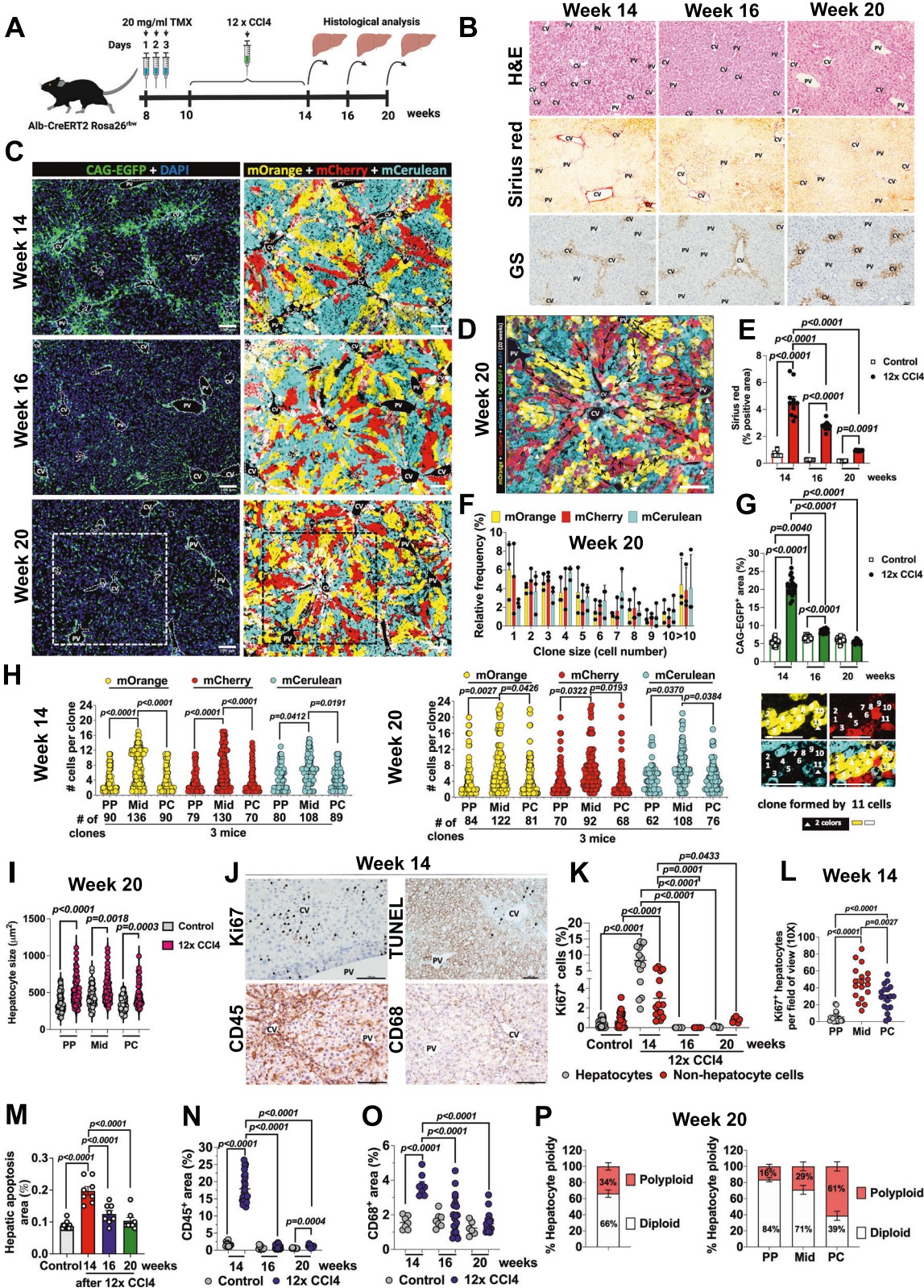

treatment with CCl4 (Fig. 3M). We found that most TUNEL positive cells were hepatocytes around central veins and adjacent to Mid areas by IHC of consecutive slides at week 14 (Fig. 3J). Moreover, we also observed apoptotic non-hepatocyte cells around portal triads and central veins (Fig. 3J). In both cell-types, TUNEL staining was scarce although statistically significant compared to livers from control mice (TUNEL staining at week 14) (Fig. 3J, M and Supplementary Fig. 3G).

Apoptosis was notably reduced at the recovery time points at weeks 16 and 20, similar to values of livers from control mice (Fig. 3M). Given we detected a significant elevation of proliferating Ki67+ non-hepatocyte cells after the 12 doses of CCl4 in week 14 (Fig. 3K), we further examined the contribution of liver immune cells to this increase. We quantified CD45 and CD68 positive cells by IHC (Fig. 3J, N, O). Overall, we identified a considerable concentration of CD45+ cells around the

**Fig. 3 | Midlobular hepatocytes remodel the liver architecture after CCl4-induced chronic injury. A** Schematic showing the experimental design for the CCl4-induced chronic liver injury model. Created with BioRender.com. **B** Liver sections stained with H&E, Sirius red and GS from mice at week 14, 16 and 20 of age after receiving additional doses of CCl4. Scale bars, 100 μm. **C** Representative immunofluorescence images of liver sections from mice at 14, 16 and 20 weeks of age after CCl4 treatment ($n = 4$). Scale bars, 100 μm. **D** Detailed view of the liver lobule architecture showing the large hepatocyte clones generated during the recovery period in mice at week 20 of age, 6 weeks post-CCl4 chronic damage. Black arrows show clonal expansion originated from PP hepatocytes to PC zones. Portal tracts are identified by bile ducts (arrowhead). Red circles show isolated midlobular clones. Scale bars, 100 μm. **E** Quantification of Sirius red staining in liver sections from mice treated with vehicle or CCl4 for 4 weeks (shown in Fig. 3B) and euthanized at week 14, 16 and 20 of age. Data are presented as mean values +/− SEM. 8–5 photos per mouse from a total of 3 mice per group. One-way ANOVA, Tukey's multiple comparisons post-test. **F** Relative frequency in % of each clone size per area of the liver lobule and per Rainbow fluorophore from mice at week 20 of age, 6 weeks post-CCl4 chronic damage. Data are presented as mean values +/− SEM. 46 liver lobule areas analyzed from a total of 3 mice (5 photos per-mouse). **G** Quantification of the % for the positive area per field of view (10×) for CAG-EGFP in liver sections from vehicle and CCl4 treated mice euthanized at week 14, 16 and 20 of age. Data are presented as mean values +/− SEM. 8 photos analyzed per mouse from 3 mice per group. One-way ANOVA, Tukey's multiple comparisons post-test. **H** Quantification of number of cells per clone (y-axis) and number of clones (x-axis) per area of the liver lobule and per Rainbow fluorophore from mice treated with CCl4 for 4 weeks and euthanized at week 14 and 20 of age. Data are presented as mean values. 31–46 liver lobule areas analyzed from 3 mice per group (4–5 photos per-mouse). One-way ANOVA, Tukey's multiple comparisons post-test. Right image: Immunofluorescence staining from mice at week 20 of age showing a large hepatocyte clone formed by 11 cells. 10 cells expressed only one color indicating that most cells of the clones generated during the recovery period after CCl4 were diploid. Scale bars, 100 μm. **I** Hepatocyte size of clones developed within the liver lobule during the recovery period at 20 weeks of age from mice chronically treated with vehicle or CCl4. Data are presented as mean values +/− SEM. 52–63 liver lobule areas analyzed from 3 mice per condition (5 photos per mouse). One-way ANOVA, Tukey's multiple comparisons post-test. **J** Images of Ki67, CD45 and CD68 IHC and TUNEL assay in liver sections from mice treated with CCl4 for 4 weeks and euthanized at week 14. Scale bars, 100 μm. **K** Quantification of total Ki67 positive cells (hepatocytes and non-hepatocyte cells) in % per field of view (10×) of liver sections from mice treated with vehicle or CCl4 for 4 weeks and euthanized at week 14, 16 and 20 of age. Data are presented as mean values +/− SEM. 23–40 photos analyzed from 3 mice per group. One-way ANOVA, Tukey's multiple comparisons post-test. **L** Quantification of proliferating Ki67 hepatocytes per field of view (10x) in each zone of the liver lobule from mice treated with CCl4 for 4 weeks and euthanized at week 14. Data are presented as mean values +/− SEM. 18–20 photos analyzed from 3 mice. One-way ANOVA, Tukey's multiple comparisons post-test. **M** Quantification of TUNEL staining. Apoptosis area in % in liver sections from untreated control and CCl4 treated mice euthanized at weeks 14, 16 and 20 of age. Data are presented as mean values +/− SEM. 7–12 photos analyzed per mouse; 3 mice per group. One-way ANOVA, Tukey's multiple comparisons post-test. **N** Quantification of CD45 IHC. CD45 area in % per field of view (10×) in liver sections from untreated control and CCl4 treated mice euthanized at weeks 14, 16 and 20 of age. Data are presented as mean values +/− SEM. 10–21 photos analyzed per mouse; 3 mice per group. One-way ANOVA, Tukey's multiple comparisons post-test. **O** Quantification of CD68 IHC. CD68 area in % per field of view (10×) in liver sections from untreated control and CCl4 treated mice euthanized at weeks 14, 16 and 20 of age. Data are presented as mean values +/− SEM. 8–15 photos analyzed per mouse; 3 mice per group. One-way ANOVA, Tukey's multiple comparisons post-test. **P** Hepatocyte ploidy in the liver lobule of mice at 20 weeks of age or 6 weeks post-CCl4 chronic damage. % of total polyploid and diploid hepatocytes within the liver lobule (left graph) and per area of the liver lobule (right graph). Data are presented as mean values +/− SEM. 5 photos analyzed per mouse; $n = 3$ mice. Source data are provided as a Source Data file.

central veins at week 14 (Fig. 3J), which rapidly dropped during the recovery periods of weeks 16 and 20, like those counted in livers from control mice (Fig. 3N and Supplementary Fig. 3G). The pattern for CD68$^+$ cells was similar to that of CD45$^+$ staining but it was less intense (Fig. 3J, O and Supplementary Fig. 3G).

## DDC injury-induced ductular reaction causes proliferation of midlobular hepatocytes

We next used the DDC diet to induce both biliary and hepatic lineage damages and evaluate potential mechanisms of differentiation, dedifferentiation and transdifferentiation in vivo. To track the liver cells, we injected 8-week-old *Alb-CreERT2 Rosa26$^{rbw}$* mice with TMX and, after a 2-week washout period we fed the mice with a chow or a 0.1% (w/w) DDC diet for 6 weeks (Fig. 4A). We then sacrificed the mice and measured body, liver and spleen weights (Fig. 4C). Although the DDC diet was well tolerated, mice exhibited a significant reduction in their body weights compared to mice fed a chow diet (Fig. 4C). DDC treatment did not alter the spleen weight (Fig. 4C); however, it did cause significant hepatomegaly illustrated by a notable increase in liver weight (Fig. 4C). To deeply characterize this type of lesion, we histologically analyzed the liver tissues from both diet interventions (Fig. 4B–N). As previously described, after the DDC treatment, mice developed cholangitis evidenced by the presence of biliary porphyrin deposition and periductular fibrosis in H&E and Sirius Red staining respectively (Fig. 4B, G). Immunofluorescence analysis revealed that the DDC diet caused the death of almost all PP hepatocyte population (Fig. 4G). However, some individual hepatocytes surrounded by the bile ducts preserved cytoplasmatic Rainbow fluorescence, and normal non-damaged nucleus (Fig. 4G). Increased cellular apoptosis was validated by the increased positive TUNEL staining around portal veins in these mice (Fig. 4G, I, L). However, no apoptotic cells were found in livers from control mice (Fig. 4G, I, L). In addition, liver sections of these mice showed a significant elevation in the number of total cells

represented by DAPI staining (Fig. 4D) and % of the non-hepatocyte area as the CAG-EGFP staining exhibited compared to control mice (Fig. 4E). We confirmed by IHC of consecutive sections that a prominent CAG-EGFP$^+$ cell population corresponded to biliary cells using the ductal marker CK19 (Fig. 4F). CK19 staining was overexpressed in liver sections from DDC treated mice for cells whose cytoplasm was smaller than those from hepatocytes confirming that portal triads were expanded with numerous bile ducts of poorly defined lumens (Fig. 4F). CK19 staining was exclusively expressed in cholangiocytes or biliary cells (Fig. 4F). We did not observe any hepatocyte-like cell morphology positive for the marker of non-hepatocytes CAG-EGFP (Fig. 4G). We did not see either any hepatocyte positive for both, any rainbow color (mCherry$^+$, mOrange$^+$ y/o mCerulean$^+$) and CK19 and none of the hepatocytes were arranged in a ductular configuration (Fig. 4G, F). Furthermore, our immunofluorescence analysis did not reveal any Rainbow colored biliary-like cell (Fig. 4G). Taken together, these results confirm that there is not any mechanism of transdifferentiation between biliary and hepatic epithelia as we hypothetically could have found in our multicolor lineage tracing *Alb-CreERT2 Rosa26$^{rbw}$* mice (Supplementary Fig. 4A, B). In control mice, CK19 positive staining was restricted to the cells that form the bile ducts of the portal triads (Fig. 4F).

In concurrence with previous work, these data confirmed that our model of 6-weeks of DDC injury stimulated the proliferation of CK19 positive biliary epithelial cells, forming multiple bile ducts in the portal triads, an event known as ductular reaction (DR). Given DR is associated with inflammatory cell infiltration in portal areas and the presence of numerous CK19 negative non-hepatocyte cells by CAG-EGFP, we investigated the expression of CD45 and CD68 markers after DDC feeding (Fig. 4J, M). We observed CD45$^+$ cells significantly accumulated in PP regions (Fig. 4J). This population could correspond with infiltrating lymphocytes that collaborate to the local inflammatory response induced by biliary obstruction. However, we did not observe

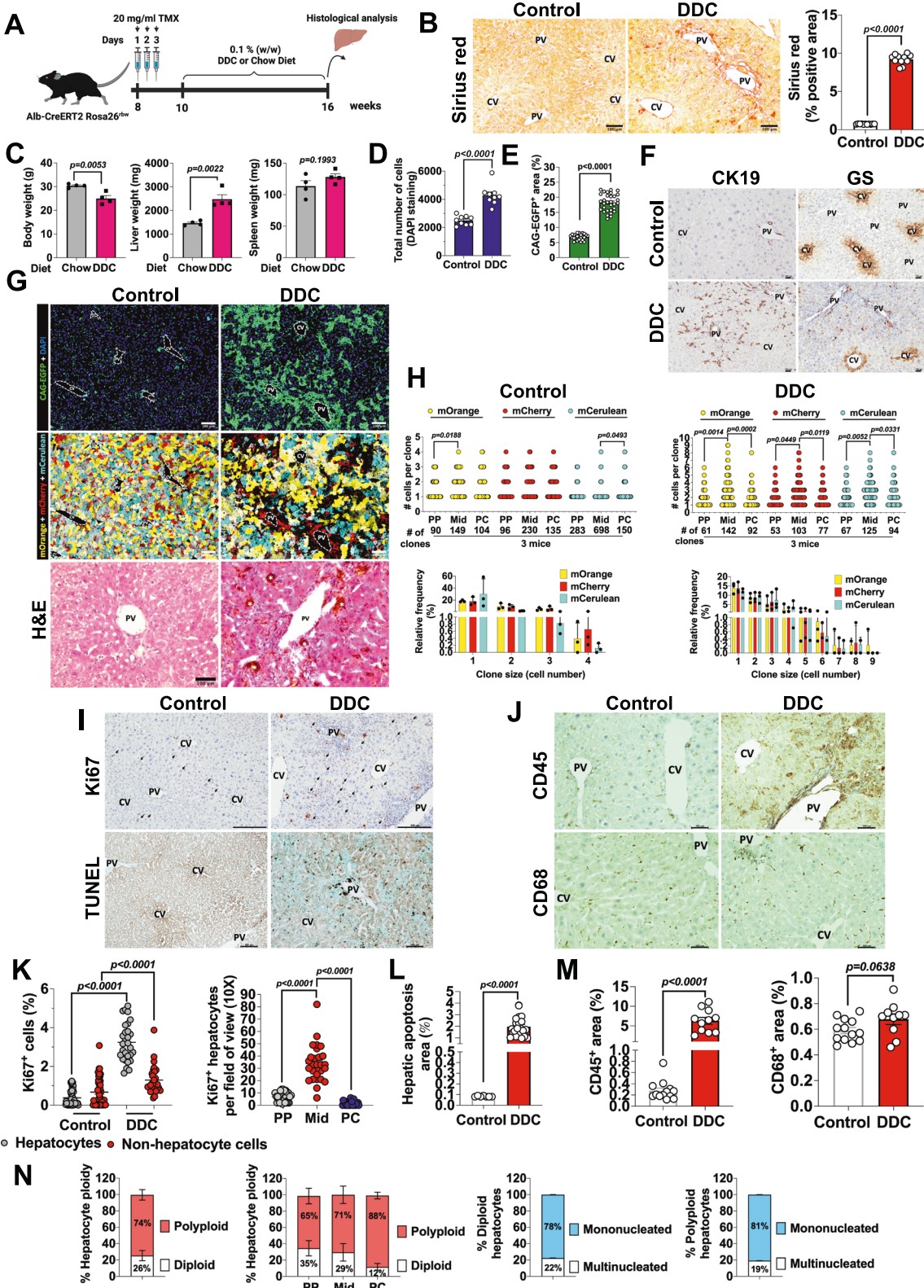

notable differences in CD68 staining between DDC treated and untreated mice (Fig. 4J, M). In addition, we readily detected clones containing 4–9 hepatocytes after DDC exposure through our immunofluorescence analysis (Fig. 4G, H). These clones were primarily located in the Mid areas of the liver lobule (Fig. 4G, H) and their cell and nuclear sizes were smaller than those from PC hepatocytes. Liver sections from control mice displayed clones formed by 1–3 hepatocytes (Fig. 4G, H), as previously observed in liver homeostasis (Fig. 1G,

H). In agreement with this data, liver lobules from mice treated with the DDC diet showed a dramatic increase in Ki67 Mid hepatocyte staining compared to their PP or PC neighbor populations (Fig. 4I, K). GS expression, considered a marker of PC hepatocytes, was reduced in mice fed a DDC diet, with the most GS positive staining concentrated, in the first layer of PC hepatocytes (Fig. 4F). Meanwhile, livers from control mice showed an expanded GS expression though 1–3 layers of hepatocytes (Fig. 4F). These observations suggest DDC injury also

**Fig. 4 | DDC injury-induced ductular reaction causes proliferation of mid-lobular hepatocytes. A** Schematic showing the experimental design for the DDC-induced bile duct and hepatocyte injury model. Created with BioRender.com. **B** Images and quantification of Sirius red staining in liver sections from mice fed chow (control mice) or DDC diet for 6 weeks. Data are presented as mean values +/− SEM. 11 photos from 3 mice per group. Unpaired $t$ test. CV central vein, PV portal vein. Scale bars, 100 μm. **C** Body weight (g), liver weight (mg) and spleen weight (mg) from mice fed chow or DDC diet for 6 weeks; Data are presented as mean values +/− SEM; $n = 4$ mice per diet. Unpaired $t$ test. **D** Quantification of total number of cells by DAPI staining in mice fed chow or DDC diet for 6 weeks. Data are presented as mean values +/− SEM. 9 photos analyzed from 3 mice per group. Unpaired $t$ test. **E** Quantification of the % for the positive area per field of view (10×) for CAG-EGFP in liver sections from mice fed chow and DDC diet for 6 weeks. Data are presented as mean values +/− SEM. 5–14 photos analyzed per mouse; $n = 3$ mice per group. Unpaired $t$ test. **F** Representative CK19 and GS IHC analysis in liver sections from mice fed chow or DDC diet for 6 weeks ($n = 4$). Scale bars, 100 μm. **G** Liver sections from mice fed chow (left panels) or DDC diet (right panels) for 6 weeks. Top panels show CAG-EGFP and DAPI staining. Middle panels indicate the Rainbow fluorophores. Lower panels represent H&E staining showing a portal vein from both groups of mice. Asterisks indicate biliary porphyrin deposition. Scale bars, 100 μm. **H** Top graphs: quantification of number of cells per clone ($y$-axis) and number of clones ($x$-axis) per area of the liver lobule and per Rainbow fluorophore of liver sections from mice fed chow (control) or DDC diet for 6 weeks. Data are presented as mean values. 22–50 liver lobule areas analyzed from 3 mice per group (7–10 photos per mouse). One-way ANOVA, Tukey's multiple comparisons post-test. Bottom graphs: relative frequency in % of each clone size per area of the liver lobule and per Rainbow fluorophore from mice fed chow (control) or DDC diet for 6 weeks. Data are presented as mean values +/− SEM. 22–50 liver lobule areas analyzed from 3 mice per group (7–10 photos per mouse). **I** Ki67 IHC and TUNEL assay in liver sections from mice fed chow (control) or DDC diet for 6 weeks. Scale bars, 100 μm. **J** CD68 and CD45 IHC in liver sections from mice fed chow (control) or DDC diet for 6 weeks. Scale bars, 100 μm. **K** Quantification of the Ki67 IHC. Left graph: Total Ki67 positive cells (hepatocytes and non-hepatocyte cells) in % per field of view (10×) of liver sections from mice fed chow (control) or DDC diet for 6 weeks. 20 and 30 photos analyzed from 3 mice for control and DDC mice respectively. Right graph: proliferating Ki67 hepatocytes per field of view (10×) in each zone of the liver lobule from mice fed chow (control) or DDC diet for 6 weeks. 20 and 30 photos analyzed from 3 mice for control and DDC mice respectively. Data in left and right panels are presented as mean values +/− SEM. One-way ANOVA, Tukey's multiple comparisons post-test. **L** Quantification of the TUNEL positive staining. Apoptosis area in % in liver sections from mice fed chow (control) or DDC diet for 6 weeks. 7–18 photos analyzed per mouse; 3 mice per group. Data are presented as mean values +/− SEM. Unpaired $t$ test. **M** Quantification of the CD68 and CD45 positive staining. CD68 and CD45 area in % per field of view (10×) in liver sections from mice fed chow (control) or DDC diet for 6 weeks. CD68: Data are presented as mean values +/− SEM. 12–13 photos analyzed per mouse; 3 mice per group. Unpaired $t$ test. CD45: Data are presented as mean values +/− SEM. 12–14 photos analyzed per mouse; 3 mice per group. Unpaired $t$ test. **N** Hepatocyte ploidy in the liver lobule of mice fed DDC diet for 6 weeks. % of total polyploid and diploid hepatocytes within the liver lobule (left graph) and per area of the liver lobule (middle left graph). % of diploid hepatocytes (hepatocytes that express only 1 fluorophore) being mononucleated or multinucleated (middle right graph). % of polyploid hepatocytes (hepatocytes that express more than 1 fluorophore) being mononucleated or multinucleated (right graph). Data are presented as mean values +/− SEM. 5 photos analyzed per mouse; 3 mice. Source data are provided as a Source Data file.

damaged the functionality of PC hepatocytes, reducing the expression of key metabolic enzymes, although not their proliferative potential, given we observed some Ki67 positive staining in PC hepatocytes (Fig. 4I). Finally, we characterized the hepatic ploidy in each zone of the liver lobule after 6 weeks of DDC diet to evaluate whether new hepatocytes generated in Mid areas differed in ploidy from control mice (Figs. 4N and 1J). We found 74% polyploid and 26% diploid hepatocytes within the liver lobule (Fig. 4N), similar values to those found in liver homeostasis (Fig. 1J). Like in control mice, Mid and PC zones resulted in having a majority of polyploid hepatocytes, content with a 71% and 88% respectively compared to PP areas, where 65% of the hepatocytes were polyploid. However, in this damage model, PP was mostly diploid (35%), followed by Mid (29%) and PC (12%) (Fig. 4N). Although 78% of diploid hepatocytes were mononucleated, 81% of polyploid hepatocytes turned out to also be mononucleated (Fig. 4N). So, with this model of cholestatic liver damage we confirm our previous findings in healthy and CCl4 injury livers: that nuclear ploidy does not match hepatocyte ploidy (Fig. 4N).

## Midlobular hepatocytes repopulate the liver during regeneration after 2/3 PHx

We also explored the remarkable ability of the liver to regenerate itself by examining the contribution of different hepatocyte populations in restoring liver lobule mass. To this end, we subjected TMX-treated *Alb-CreERT2 Rosa26^{rbw}* mice to 2/3 PHx and we traced hepatocyte clonal dynamics after physical injury. Mice were sacrificed at 24, 48, 72 and 162 h (or 7 days) after PHx ($n = 2$ per group) which encompassed the main regenerative burst of PHx (Fig. 5A). The removed liver lobes were used as hour 0 controls. Weights from the whole body, liver and spleen, as well as liver/body weight ratio were measured for each time point (Fig. 5B). As expected, liver weight decreased at 24 h, increased at 48 h and almost regained its initial weight at 72 h after resection. After this latter time point, liver weight remained constant until 162 h (Fig. 5B). Mouse body weight dropped at 24 and 48 h after PHx. From this point on, body weight began to increase with a slight recovery at 72 h maintained until 162 h post-resection. The fast rebound in liver weight and continued loss of body weight at 48 h

culminated in rapid restoration of the liver/body weight ratio, which was almost completely reestablished by 48–72 h. This ratio was restored even though liver and whole-body weights did not reach their initial values within the time points studied (Fig. 5B). Following PHx, spleen weight gradually increased over time (Fig. 5B). We also investigated the clonal expansion of hepatocytes in the regenerating liver (Fig. 5C–E). At time 0 h, most of the hepatocytes were individual cells or clones formed by 2 or 3 hepatocytes although a few clones formed by 4 hepatocytes were also observed (Supplementary Fig. 5A). These clones were more numerous in the Mid area compared to PP or PC areas (Supplementary Fig. 5A). These findings were similar to those which we observed previously during liver homeostasis (Fig. 1G, H). After 24 h of PHx, individual hepatocytes were less common that at 0 h (Fig. 5C–E). The decrease in individual hepatocytes was concurrent with an increase in the relative frequency of clones formed by 2, 3 and 4 hepatocytes, principally this last one (Fig. 5C–E). Midlobular zones were where we mainly observed more numerous and larger clones (Fig. 5C–E). This clonal expansion continued through 48 h after the resection when the number and size of clones increased considerably, appearing clones of 5 hepatocytes. This dynamic was decreased steadily over the 72 and 162 h time points (Fig. 5C–E). Similarly, to previous time points, Mid hepatocytes were more likely to be larger and more numerous than PP or PC hepatocytes at 48, 72 and 162 h after the injury (Fig. 5C, D). We also examined hepatocyte proliferation by IHC for Ki67 on consecutive liver sections (Fig. 5G, Supplementary Fig. 5A,C). Our results showed Ki67 proliferating hepatocytes were significantly increased at 24 and 48 h after the injury (Fig. 5G and Supplementary Fig. 5C). This maintained hepatocyte proliferative wave tapered down through the 72 and 162 time points (Fig. 5G and Supplementary Fig. 5C). Proliferating hepatocytes were reduced at 72 h, followed by 162 h (Fig. 5G and Supplementary Fig. 5C). We found that Mid hepatocytes were the most proliferative at 24, 48 and 72 h after PHx (Fig. 5H). Taken together, these results indicate that Mid hepatocytes were the primary source of hepatocyte progeny in the regenerating liver. Beyond hepatocyte hyperplasia, we also investigated hypertrophy measuring hepatocyte size during liver regeneration. We notably found increased hepatocyte size at 48 h after the injury

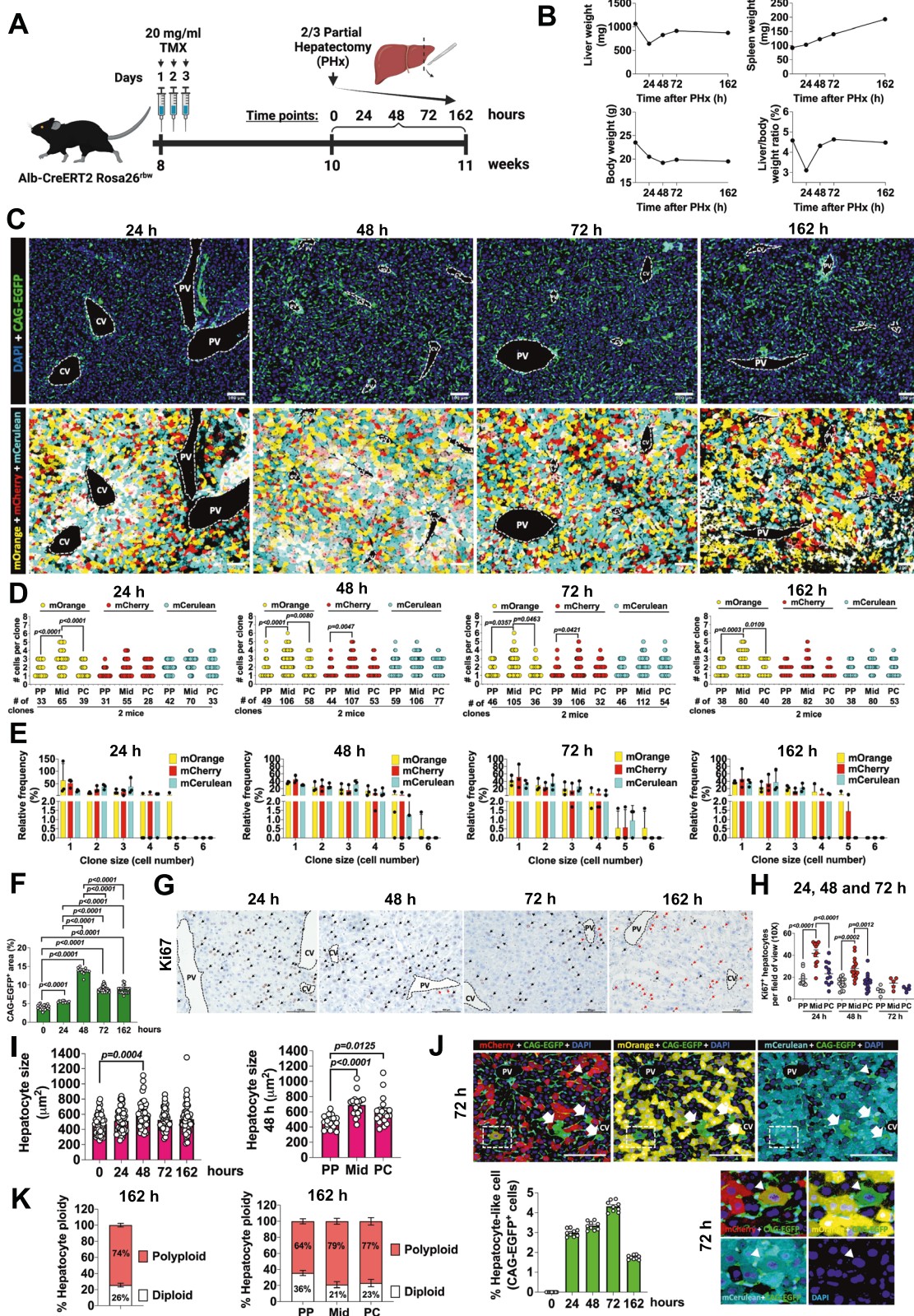

compared to time point 0 h (Fig. 5I). The largest hepatocytes were in the Mid region, followed by the PC zone (Fig. 5I).

Quantitative analysis of CAG-EGFP non-hepatocyte cells spiked at 48 h after the injury, followed by a second peak at 72 h (Fig. 5F). As liver regeneration is remarkably accompanied by intrahepatic angiogenesis[3], we quantified the marker of endothelial cells CD31 by IHC to evaluate tissue repair after PHx (Supplementary Fig. 5D). We

observed an enrichment of CD31+ cells in the regenerating liver during this interval, being in line with our CAG-EGFP staining (Fig. 5F and Supplementary Fig. 5D). Intriguingly, CAG-EGFP immunofluorescence showed some hepatocyte-like cells during liver regeneration at 24, 48, 72 and 162 h after PHx (Fig. 5J and Supplementary Fig. 5E). This hepatocyte-like cell group was first observed at 24 h and increased in number at 48 h and 72 h (Fig. 5J and Supplementary Fig. 5E). Primarily

**Fig. 5 | Midlobular hepatocytes repopulate the liver during regeneration after 2/3 PHx. A** Schematic showing the experimental design for the evaluation of liver regeneration after a 2/3 partial hepatectomy model. Created with BioRender.com. **B** Liver (mg), spleen (mg) and body weights (g) and liver/body weight ratio (%) at time points 0, 24, 48, 72 and 162 h of mice sujected to 2/3 PHx. $N = 2$ mice used per time point. Data are presented as mean values. **C** Liver sections from mice subjected to 2/3 PHx after 24, 48, 72 and 162 h. Scale bars, 100 μm. **D** Quantification of number of cells per clone (*y*-axis) and number of clones (*x*-axis) per area of the liver lobule and per Rainbow fluorophore from mice subjected to 2/3 PHx after 24, 48, 72 and 162 h. Data are presented as mean values. 11–12 liver lobule areas analyzed from 2 mice per group (4 photos per-mouse). One-way ANOVA, Tukey's multiple comparisons post-test. **E** Relative frequency in % of each clone size per area of the liver lobule and per Rainbow fluorophore from mice subjected to 2/3 PHx after 24, 48, 72 and 162 h. Data are presented as mean values. 11–12 liver lobule areas analyzed from 2 mice per group (4 photos per-mouse). **F** Quantification of the % for the positive area per field of view (10×) for CAG-EGFP in liver sections from mice subjected to 2/3 PHx at time points 0, 24, 48, 72 and 162 h. Data are presented as mean values +/− SEM. 7–18 photos per mouse; 2 mice per group. One-way ANOVA, Tukey's multiple comparisons post-test. **G** Ki67 IHC in liver sections from mice subjected to 2/3 PHx after 24, 48, 72 and 162 h. Arrows indicate proliferating Ki67 hepatocytes. $N = 2$ mice. Scale bars, 100 μm. **H** Quantification of proliferating Ki67 hepatocytes per field of view (10×) in each zone of the liver lobule from mice subjected to 2/3 PHx

after 24, 48 and 72 h. Data are presented as mean values +/− SEM. 10–15 liver lobule areas analyzed from 2 mice per group. One-way ANOVA, Tukey's multiple comparisons post-test. **I** Left: Hepatocyte size during liver regeneration in liver sections from mice subjected to 2/3 PHx at time points 0, 24, 48, 72 and 162 h. 9–12 liver lobule areas analyzed from 2 mice. Right: Hepatocyte size within the liver lobule from mice 48 h after being subjected to 2/3 PHx. 9 liver lobule areas analyzed from 2 mice. Data in left and right panels are presented as mean values +/− SEM. One-way ANOVA, Tukey's multiple comparisons post-test. **J** Images after 72 h of PHx and quantification of % CAG-EGFP positive hepatocyte cells-like of liver sections from mice subjected to 2/3 PHx. Arrowheads indicate hepatocyte cell-like positive for both fluorophores, the non-hepatocye marker CAG-EGFP and the hepatocyte marker mOrange. Arrows indicate hepatocyte cell-like positive for the non-hepatocye marker CAG-EGFP. Dashed boxes show a detail view of a hepatocyte cell-like (arrow) positive for CAG-EGFP, mCherry and mCerulean. Lower left graph represents the % of CAG-EGFP positive hepatocyte cells-like per field view (10×) of liver sections from mice subjected to 2/3 PHx at time points 0, 24, 48, 72 and 162 h. Data are presented as mean values +/− SEM. Scale bars, 100 μm. **K** Hepatocyte ploidy in the regenerated liver from mice after 162 h of being subjected to 2/3 PHx. % of total polyploid and diploid hepatocytes within the liver lobule (left graph) and per area of the liver lobule (right graph). Data are presented as mean values +/− SEM. 5 photos analyzed per mouse, 2 mice per group. Source data are provided as a Source Data file.

located in the Mid area of the regenerating liver lobules, this CAG-EGFP$^+$ cells apparated not just in isolation, but some were in clonal clusters of 2–6 cells (Fig. 5J). These cells were still present at 162 h after the injury and were not observed in the resected tissue at 0 h (Fig. 5J and Supplementary Fig. 5E). The presence of this type of hepatocyte-like cell population during liver regeneration may suggest the existence of transdifferentiation processes from non-hepatocyte cells into hepatocytes. These cells resulted negative for the biliary epithelial cells and liver progenitor cells marker CK19, which was strictly expressed by bile duct cells in the portal areas (Supplementary Fig. 5F). Unexpectedly, we also observed some of these hepatocyte-like cells positive for both markers, CAG-EGFP and ≥1 Rainbow fluorophores (mOrange, mCherry or/and mCerulean) (Fig. 5J). Most of these marker-overlapping hepatocyte like-cells were multinucleated (Fig. 5J). Cell fusion, described in the liver[22] could explain the origin of this type of unusual hepatocyte-like cells. In addition, given that some authors have documented that PHx conducted in rats increased multi-nucleated polyploid hepatocytes, we investigated changes in hepatocyte ploidy[23]. We found no alteration on liver ploidy after PHx, with 74% polyploid and 26% diploid hepatocytes within the liver lobule (Fig. 5K), similar values to those found in liver homeostasis (Fig. 1J). Like in control mice, Mid and PC zones had a similar majority of polyploid hepatocytes, 79% and 77% respectively, compared to PP areas, where 64% of the hepatocytes were polyploid (Fig. 5K). We did not observe elevation in the number of multinucleated polyploid hepatocytes, in fact, we found the opposite, with 95% of polyploid hepatocytes being mononucleated (Supplementary Fig. 5G). We confirm with these results our previous finding in the healthy and injured livers: that nuclear ploidy does not match hepatocyte ploidy (Supplementary Fig. 5G).

## Hyperplasia and hypertrophy of midlobular hepatocytes repair the liver after coronavirus injury

Severe acute respiratory syndrome coronavirus 2 (SARS-CoV-2), which caused the coronavirus disease 2019 (COVID-19) pandemic, targets mainly the lung due to the abundant expression of its receptor, angiotensin-converting enzyme 2 (ACE2), by respiratory epithelium. However, COVID-19 patients are also at risk of liver damage[24,25]. This finding is in line with studies that found high ACE2 receptor expression in the intrahepatic bile ducts[26] and SARS-CoV-2 particles in the cytoplasm of hepatocytes from liver biopsies of COVID-19 individuals[27]. These data indicate that SARS-CoV-2 is capable of directly injuring the

liver. The cytotoxicity of infection in the liver could lead to a systemic inflammatory response followed by multiorgan failure, which is observed in patients with severe COVID-19[28]. As SARS-CoV-2 is highly phylogenetically related to the murine native beta coronavirus Mouse Hepatitis Virus (MHV) Strain A59[29], MHV-A59 has been used as a pre-clinical coronavirus model to reproduce many aspects of COVID-19 pathophysiology in mice[30]. Like SARS-COV-2, intranasal inoculation of MHV-A59 in mice produces acute respiratory distress syndrome, lymphopenia, multiorgan involvement, and systemic inflammation[30]. MHV-A59 also targets the liver via the CEACAM1 receptor, which, like the ACE2 receptor, is highly expressed in this tissue and, is also interferon-inducible[31,32].

We used the MHV-A59 model to explore the pathological changes in the biliary cells and hepatocytes triggered by coronavirus infection and the regenerative capacity after this viral injury. We intranasally infected 10-week-old male *Alb-CreERT2 Rosa26$^{rbw}$* mice with MHV-A59 after activating the Rainbow fluorophores with TMX. Mice sacrificed at 10 weeks of age were our control group and were considered week 0. At 1- and 4-weeks post-infection, we sacrificed the mice to assess liver architecture after viral damage and tissue repair respectively (Fig. 6A). We then used histological techniques to investigate hepatocyte clonal expansion, liver morphology, hyperplasia, hypertrophy and apoptosis (Fig. 6 and Supplementary Fig. 6). We used males in our study, as females were shown to be significantly more resistant to MHV-A59 and COVID-19 infection[30,33,34]. From days 1–7 after infection, the animals in the infected groups demonstrated respiratory illness symptoms, such as fever, shortness of breath, and a gradual decrease in body weight from days 0–7 compared with to PBS treated controls, which appeared normal over the entire period (Fig. 6B).

We first examined liver changes by immunofluorescence after MHV-A59 infection (Fig. 6C). We identified a significant increase in total cells at 1- and 4-weeks post-infection compared to control mice by quantifying the number of DAPI stained nuclei (Fig. 6D). In addition, we detected more total cells at 1 week than 4 weeks post-infection (Fig. 6D). Similarly, we found higher CAG-EGFP$^+$ area (%) at 1 week than 4 weeks after infection, indicating the generation and elimination of non-hepatocyte populations respectively (Fig. 6E). We also observed hepatocyte clonal expansion at 4 weeks post-infection compared to 1 week, indicative of liver repair processes (Fig. 6F, G). Particularly, hepatocyte clones within the liver lobule increased in number and size (Fig. 6F, G). Larger clones of 4 to 7 cells were detected at 4 weeks post-infection and their relative frequency was higher than at 1-week post-

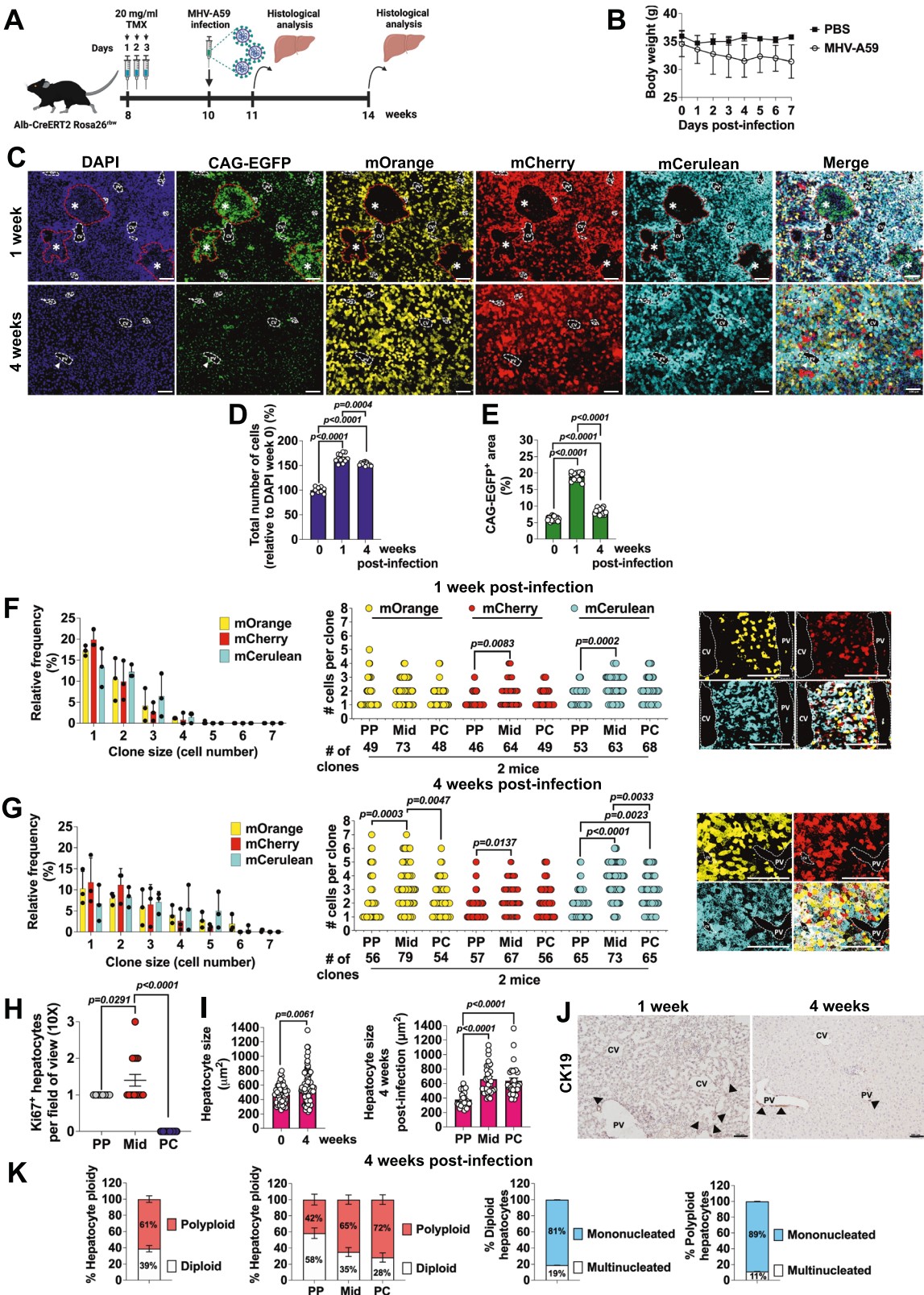

infection (Fig. 6F, G). These larger clonal clusters were in the Mid areas, followed by PC and PP zones (Fig. 6F, G). These findings were consistent with our observations in other liver injury models, such as, CCl4 damage and PHx. In line with this observation, we also observed more proliferating Ki67 hepatocytes in the Mid areas at week 4 post-infection (Fig. 6H). Liver regeneration after 4 weeks was also accompanied by hepatocyte hypertrophy, with Mid hepatocytes being the largest (Fig. 6I). On week 1 post-infection, H&E staining and TUNNEL assay demonstrated massive hepatic necrosis and apoptosis respectively (Supplementary Fig. 6A–B). Immunohistochemical analysis of CD68 and CD45 markers detected considerable inflammatory infiltrates mainly distributed within the liver parenchyma, from central veins to Mid areas of the liver lobule (Supplementary Fig. 6D–E). Although, we also found CD68 and CD45 expressed by non-

**Fig. 6 | Hyperplasia and hypertrophy of midlobular hepatocytes repair the liver after coronavirus injury. A** Schematic showing the experimental design for coronavirus liver injury with MHV-A59. Created with BioRender.com. **B** Body weight curves in 10-week-old male mice during the first week of viral infection with MHV-A59 or PBS vehicle treatment; n = 3–4 mice per group. Data are presented as mean values +/− SEM. **C** Liver sections from mice after 1 and 4 weeks of MHV-A59 viral infection. Asteriks indicate immune cell infiltrates. CV central vein, PV portal vein. Scale bars, 100 µm. **D** Quantification of DAPI staining or total number of cells per field of view (10×) in liver sections from mice before and after 1 and 4 weeks of MHV-A59 infection. Data are presented as mean values +/− SEM. 5–8 photos analyzed per mouse; n = 3 mice per group. One-way ANOVA, Tukey's multiple comparisons post-test. **E** Quantification of CAG-EGFP staining per field of view (10×) in liver sections from mice 1 and 4 weeks post-MHV-A59 infection. Data are presented as mean values +/− SEM. 5–8 photos analyzed per mouse; n = 3 mice per group. One-way ANOVA, Tukey's multiple comparisons post-test. **F** Hepatocyte injury 1 week post-MHV-A59 infection in livers from mice. Left: Relative frequency in % of each clone size per area of the liver lobule and per Rainbow fluorophore from mice at 1 week post-MHV-A59 infection. Data are presented as mean values of the average PP, Mid and PC hepatocyte population +/− SEM. 6 liver lobule areas analyzed per mouse; 2 mice per time-point (4–5 photos per mouse). Middle: quantification of number of cells per clone (y-axis) and number of clones (x-axis) per area of the liver lobule and per Rainbow fluorophore from mice at 1 week post-MHV-A59 infection. Data are presented as mean values. 6 liver lobule areas analyzed from 2 mice per time point (4–5 photos per mouse). One-way ANOVA, Tukey's multiple comparisons post-test. Right: Representative images of a liver lobule from mice at 1 week post-MHV-A59 infection; n = 2 mice. **G** Clonal expansion of hepatocytes 4 weeks post-MHV-A59 infection in livers from mice. Left: Relative frequency in % of each clone size per area of the liver lobule and per Rainbow fluorophore from mice at 4 weeks post-MHV-A59 infection. Data are presented as mean values of the average PP, Mid and PC hepatocyte population +/− SEM. 6 liver lobule areas analyzed per mouse; 2 mice per time-point (4–5 photos per mouse). Middle: quantification of number of cells per clone (y-axis) and number of clones (x-axis) per area of the liver lobule and per Rainbow fluorophore from mice at 4 weeks post-MHV-A59 infection. Data are presented as mean values. 6 liver lobule areas analyzed per mouse; 2 mice per time point (4–5 photos per mouse). One-way ANOVA, Tukey's multiple comparisons post-test. Right: Representative images of a liver lobule from mice at 4 weeks post-MHV-A59 infection; n = 2 mice. Scale bars, 100 µm. **H** Quantification of proliferating Ki67 hepatocytes per field of view (10×) in each zone of the liver lobule from mice at week 4 post-MHV-A59 infection; Data are presented as mean values +/− SEM. 9–15 liver lobule areas analyzed from 2 mice per time point. One-way ANOVA, Tukey's multiple comparisons post-test. **I** Hepatocyte hypertrophy after 4 weeks of MHV-A59 infection. Left: Hepatocyte size after tissue repair in liver sections from mice 4 weeks after MHV-A59 infection. Data are presented as mean values +/− SEM. Hepatocyte size (n = 53–91), from 3–4 liver lobule areas were analyzed from 2 mice. Unpaired Student's t-test. Right: Hepatocyte size (n = 29) within the liver lobule from mice at 4 weeks after MHV-A59 infection. Data are presented as mean values +/− SEM. 3–4 liver lobule areas analyzed from 2 mice. One-way ANOVA, Tukey's multiple comparisons post-test. (**J**) Representative CK19 IHC in liver sections from mice at 1 and 4 weeks after MHV-A59 infection. Arrows indicate cholangiocytes forming bile ducts around the portal triads; n = 2 mice. **K** Hepatocyte ploidy in the liver lobule of mice at 4 weeks after MHV-A59 infection. % of total polyploid and diploid hepatocytes within the liver lobule (left graph) and per area of the liver lobule (middle left graph). % of diploid hepatocytes (hepatocytes that express only 1 fluorophore) being mononucleated or multinucleated (middle right graph). % of polyploid hepatocytes (hepatocytes that express more than 1 fluorophore) being mononucleated or multinucleated (right graph). Data are presented as mean values +/− SEM. 5 photos analyzed per mouse; 2 mice per group. Source data are provided as a Source Data file.

parenchymal cells, in the hepatic sinusoids, suggesting activation of Kupffer cell and immune response. Immunohistochemical staining of CK19 showed no signs of cholestasis nor ductular reaction originated by cholangiocytes (Fig. 6J). Ki67 staining of consecutive liver slides showed that cell proliferation was restricted to immune cells (Supplementary Fig. 6C). On week 4 post-infection, liver lobule architecture did not show any significant histological alteration with respect to the control group indicating that the mouse adult liver preserves its regenerative capacity after coronavirus infection (Fig. 6C and Supplementary Fig. 6). Finally, we quantified the ploidy status of hepatocytes in each zone of the liver lobule after 4-weeks of MHV-A59 infection to evaluate whether hepatocytes developed during the recovery phase underwent ploidy changes (Fig. 6K). We found an increase in diploid hepatocytes with respect to liver homeostasis (Fig. 1J), with 39% being diploid *vs.* 61% polyploid (Fig. 6K). Like in control mice, Mid and PC zones had a majority of polyploid hepatocytes, with a 65% and 72% respectively compared to PP areas, where 42% of the hepatocytes were polyploid. Contrastingly, PP were mostly diploid (58%), followed by Mid (35%) and PC (28%) (Fig. 6K). Most hepatocytes were mononucleated regardless of ploidy status, with 81% and 89% of polyploid hepatocytes being mononucleated (Fig. 6K). So, with this model of coronavirus damage, we confirm the previous findings in healthy and injured livers: that nuclear ploidy does not match hepatocyte ploidy (Fig. 6K).

## Cell–cell communication analysis reveals that Gal-9-CD44 pathway may play a critical role in regulating liver homeostasis after injury by attenuating immune cell infiltration and stimulating hepatocyte proliferation

To highlight the underlying mechanisms mediated by immune cells, endothelial cells and cholangiocytes in orchestrating the reparative response together with hepatocytes to drive liver growth after injury, we performed an analysis of cell–cell communication from our scRNA-seq dataset after acute CCl4 treatment (Fig. 7 and Supplementary Figs. 10–12). We analyzed liver preparations from *Alb-CreERT2 Rosa26^rbw* mice on day 6 after CCl4 insult, when we observed peak

proliferation of hepatocytes (Supplementary Fig. 7D). As expected, our cell–cell communication analysis revealed increased total cell–cell communication in CCl4-treated mice, however, the hepatocyte communication with other cells was surprisingly reduced by injury (Fig. 7B and Supplementary Fig. 11A).

Interestingly, we identified cholangiocytes as the liver cell type in CCl4-injured mice that interacted most with themselves and other hepatic cells through expression of ligands-receptors (Fig. 7B). Particularly, cholangiocytes showed strong interactions with other cholangiocytes, followed by monocytes and endothelial cells. Monocytes also presented a notable interaction with themselves, dendritic cells and endothelial cells. Dendritic cells and endothelial cells also displayed increased interactions with other cells. By contrast, CCl4 significantly reduced ligand-receptor interactions between hepatocytes and immune cells, including monocytes and T-cells, as well as endothelial cells (Fig. 7B).

Importantly, our analysis of ligand-receptor pairs found that *Lgals9*, the gene that encodes the protein Gal-9 which is highly expressed in the liver[35], was the most upregulated gene by CCl4 in many cell types, including hepatocytes, cholangiocytes, dendritic cells, endothelial cells, macrophages, monocytes and T-cells (Fig. 7C). Several receptors or surface binding partners have been reported for Gal-9 including the adhesion molecule cluster of differentiation 44 (CD44)[35], which is also upregulated in our cell–cell interaction analysis in the same hepatic cell types where *Lgals9* was overexpressed except in hepatocytes (Fig. 7C). These results elucidate Gal-9 as a major mediator of communication between cell types to coordinate the reparative response to liver injury.

We also investigated others enriched pathways in livers from CCl4 and control mice as shown Supplementary Fig. 11B to assess the cellular and molecular mechanism that regulate liver regeneration in response to injury. We found that numerous signaling pathways were upregulated by different cell populations 6 days after acute CCl4 damage (Supplementary Fig. 11B). Most of these pathways belong to signaling cascades associated to regulate the inflammatory and immune responses (CD22[36], MHC-I[37], CD52[38], CHEMERIN[39], CD226[40],

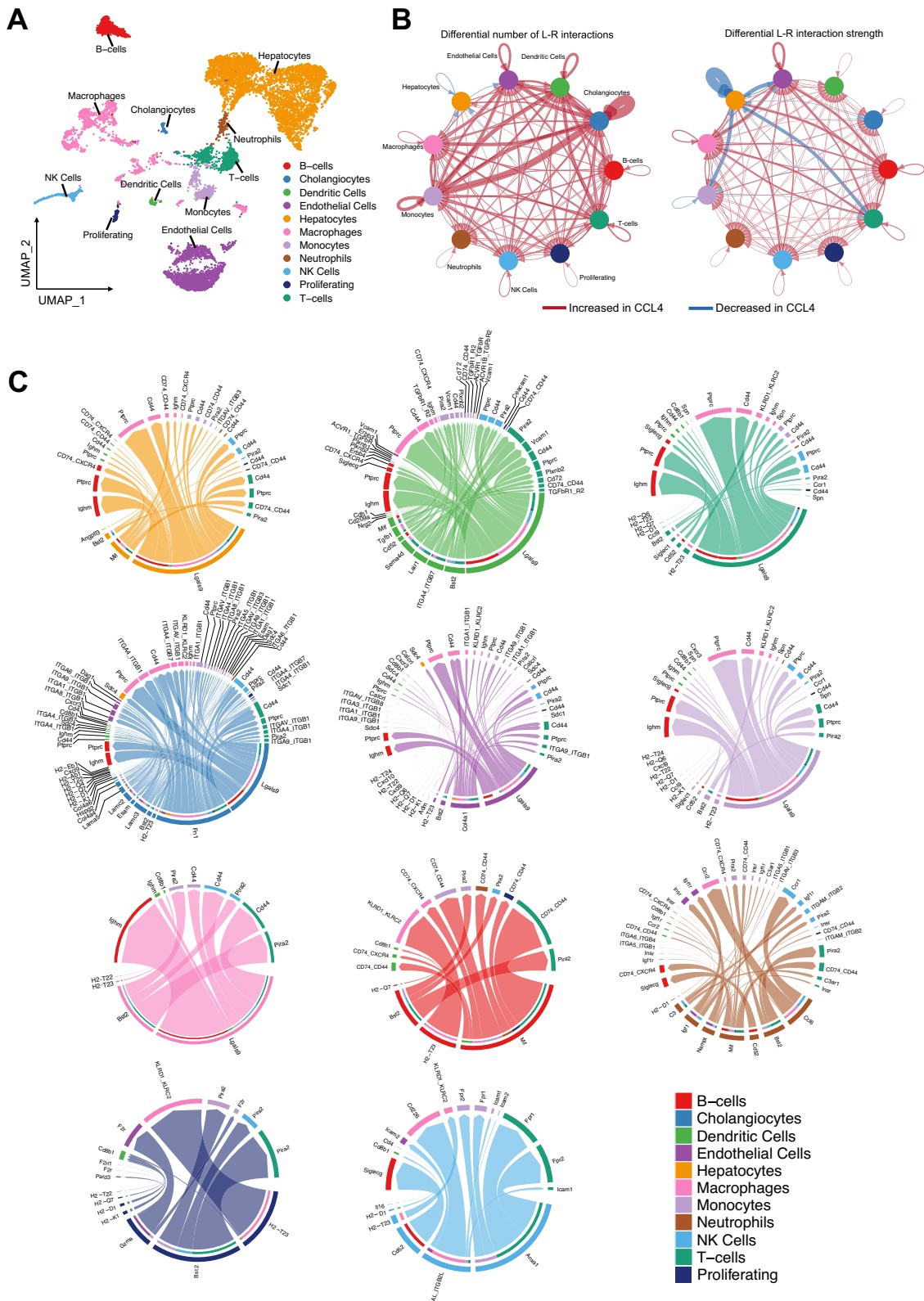

**Fig. 7 | Changes in Intracellular communications network in the liver niche during CCl4 treatment. A** UMAP clustering of single cell data with Seurat from liver preparations isolated from *Alb-CreERT2 Rosa 26*^rbw mice after 6 days of being treated with a single dose of vehicle (control) or CCl4 (CCl4). Liver cells were clustered in two dimensions using the UMAP dimensionality reduction technique and annotated by cell type. UMAP representation of 11 distinctive cell clusters from liver cells. Top 3 differentially expressed marker genes are shown for each cell cluster. *N* = 2 *Alb-CreERT2 Rosa26*^rbw mice per group. **B** Circle plot highlighting the differential number of ligand-receptor (L-R) interactions and interaction strength between control and CCl4 treated animals. *N* = 2 *Alb-CreERT2 Rosa26*^rbw mice per group. **C** Chord diagram showing upregulated signaling ligand-receptor pairs in CCl4 condition. Each link indicates a ligand-receptor pair. The root of each arrow is the ligand-expressing cell type, and the tip of each arrow is the receiving cell. *N* = 2 *Alb-CreERT2 Rosa26*^rbw mice per group.

NECTIN[41], MHC-II[42], SN, CD23[43], CD80[44], TNF[45], CSF[46], APRIL[47], IL10[48], PVR[49], AGT[50], IL16[51], CEACAM[52], CD200[53], OCLN[54], ACTIVIN[55], ANNEXIN[56], SELPLG[57], CD86[44], CXCL[58], SEMA4[59], PECAM1[60], TGFb[61], BST2[62], SELL[63], LAIR1[64], ITGAL-ITGB2[65], JAM[66], CCL[67], MIF[68], VCAM[69], ICAM[70], COMPLEMENT[71], PROS[72] and CD45[73]) (Supplementary Fig. 11B). Notably, many of the upregulated signaling pathways are known to be also activated during liver repair and regeneration (CSF[46], ACTIVIN[55], ANNEXIN[56], VCAM[69], ICAM[70], TNF[45], TGFb[61] and CEACAM[52]). Moreover, other pathways that are also drivers of hepatic remodeling resulted upregulated after CCl4 including WNT[74], HGF[75], TWEAK[76], GRN[77], NOTCH[78], IGF[79], COLLAGEN[80] and LAMININ[81] (Supplementary Fig. 11B). The upregulation of several known liver repair pathways confirms that our data set robustly assessed the response of liver to injury.

Taken together, our results indicate that CCl4 causes liver damage through free radical mediated inflammatory processes that trigger the death of PC hepatocytes initially at day 2 post CCl4, as we observed in Fig. 2C, G. This CCl4 toxicity causes macrophages and monocytes infiltration which facilitate the removal of cellular debris generated by necrosis and apoptosis of PC hepatocytes (Fig. 2H). At day 3 post CCl4, PC hepatocytes and PC clones originate by division of undamaged Mid hepatocytes, which reached a peak of hepatocyte proliferation at day 6 (Supplementary Fig. 7C, D). At this time point, cholangiocytes, monocytes and endothelial cells became very active, increasing significantly their interaction through the upregulation of numerous outgoing (ligands) and incoming (receptors) signaling patterns, as shown in greater detail in Fig. 7 and Supplementary Fig. 11C.

## Discussion

Three different hepatocyte populations have been identified along the portal-central axis of the liver lobule according to their metabolic gene pattern expression. Recent studies using single cell profiling approaches[82–84] have remarkably contributed to specifying this concept, known by the term, 'liver zonation'. This phenomenon was first revealed by histological techniques in 1944[85] and divides the liver into 3 zones. Zone 1 is formed by hepatocytes near to the portal veins named PP hepatocytes. These are mainly involved in processes such as lipid β-oxidation, cholesterol biosynthesis, ureagenesis, protein secretion and gluconeogenesis. At the other end of the lobule we find zone 3, with PC hepatocytes, which are allocated near to the central veins and execute lipogenesis, ketogenesis, glycolysis, glutamine formation, bile acid biosynthesis, and xenobiotic metabolism. Finally, the lobule is sectioned by zone 2, with Mid hepatocytes located in the transitional region between zones 1 and 3, primarily specialized in regulating iron metabolism. Besides this 'metabolic liver zonation', PP and PC hepatocytes are exposed to different gradients of oxygen and nutrients: portal areas are enriched in oxygen and nutrients, whereas pericentral regions are exposed to more hypoxia and low nutrient supplies[86]. Consequently, the zonation of gene expression in hepatocytes causes a specialization of functions[87]. This heterogeneity of PP, PC, and Mid hepatocytes is critical for the proper functioning of the whole organ in nutrient metabolism and other roles[87]. However, the proliferative gene patterns of these different hepatocyte populations were unable to conclude which area serves as a reservoir of new cells in hepatic homeostasis and regeneration for several years; significant levels of progenitor markers or central regulators of proliferative signaling pathways were found to be expressed in all 3 hepatocyte populations despite some of these regions were reported to harbor different proliferative rates[10,11,88,89].

It was not until 2021 that it became demonstrated by two independent groups that the region of the liver lobule that responds actively to loss of hepatic integrity is the Mid zone 2[8,13]. One year earlier, this liver area was also identified to play a crucial role during liver homeostasis[5]. Interestingly, one of the articles published in *Science*[13] (Wei et al.) using genetic lineage tracing approaches, scRNA-seq and CRISPR tools, found that repopulation from zone 2 may be drive by the insulin-like growth factor binding protein 2–mechanistic target of

rapamycin–cyclin D1 (IGFBP2-mTOR-CCND1) axis. These authors determined that during homeostasis, Mid zone 2 hepatocytes expanded in number without alteration of hepatocyte hypertrophy and no transdifferentiation between biliary cells and hepatocytes were observed. Mid hepatocytes also acted as the source for hepatocyte regeneration although PC or PP hepatocytes also contributed during liver remodeling, depending on which zone of the liver lobule was originally damaged. However, mechanisms of hepatocyte hypertrophy or ploidy, cell fusion or transdifferentiation were not explored during their conventional models of DDC feeding nor CCl4 insults. Similarly, the another group whose work was also published in *Science*[8], He et al. also evaluated liver homeostasis, repair, and regeneration using an elegant genetic tracing system that they developed (proliferation tracer, ProTracer) to record hepatocyte proliferation in vivo. They detected more proliferation in a subset of Mid hepatocytes during liver homeostasis, with less proliferation in PP hepatocytes and minimal proliferation in PC hepatocytes. In addition, a highly regional and dynamic hepatocyte generation pattern was observed during repair and regeneration in response to several liver injuries, with zone 2 hepatocytes mostly pioneer during the repair process, although the degree of involvement of the other hepatocyte subpopulations (PC or PP) was different between the liver damage models. No information about hepatocyte hypertrophy or ploidy nor cell fusion or transdifferentiation mechanisms were explored. These authors observed increased proliferative capacity in PP area followed by Mid in PHx, however Chembazhi et al. demonstrated that hepatocyte proliferation after PHx initiates in the Mid region before proceeding toward the PP and PC areas[14]. Taken together, the discrepancies in those findings and little or unexplored processes involved in liver regeneration and repair require a more detailed analysis. Our study compiles physical, chemical, and viral liver damage to better understand which area of the hepatic lobule harbors the greatest plastic capacity to recover liver integrity after insults of different nature. Specifically, we assessed the hepatocyte progeny after peri-central and peri-portal chemical damages, entire organ physical lesion and viral injury. Due to the uniqueness of our model, we evaluated in vivo clonal expansion, ploidy changes and potential transdifferentiation and cell fusion events that impacted hepatocyte plasticity along the hepatic lobule. Our clonal analysis in liver homeostasis and after chemical, physical and viral injuries demonstrate that mainly, proliferation of Mid hepatocytes maintains hepatocyte turnover and restores hepatic histology after injury.

In the healthy liver, we observed the largest hepatocyte clone sizes and highest hepatocyte proliferation rate in the Mid zone 2 of the lobule and the lowest in zone 3. Thus, mainly, Mid hepatocytes entered to the cell cycle to undergo division once or twice during tissue renewal. These observations agreed with previous studies that also used the Cre-loxP recombination system and identified the Mid hepatocytes as the source of new hepatocytes during liver homeostasis[8,13]. Particularly, He and colleagues traced and recorded in vivo proliferating Ki67+ cells and identified the Mid population as the principal contributor to proliferation, followed by hepatocytes located in zone 1, with modest proliferative activity, similar to our data. However, contrary to their analysis, we observed that most of the hepatocytes were individual cells rather than 2-hepatocyte size clones during liver homeostasis. This discrepancy can be attributed to differences in the periods employed between TMX-inducible Cre-activity and liver evaluations; we examined the liver after 2 weeks of TMX injection whereas He et al. employed a period of 6 weeks, giving more time for hepatocytes to undergo cell division and thus detect more than one proliferating Ki67+ hepatocytes. Notably, our findings determined that hepatocyte turnover is not equally distributed across the liver lobule or concentrated in PP or PC areas as some authors have reported previously[5,7,10]. The current study also contradicts the concept of the streaming liver, where hepatocytes progress from PV to CV during liver homeostasis, as it was firstly indicated in the literature[20,90].

We found similar outcomes in the injured liver, with the Mid population being primarily responsible for tissue repair and remodeling following liver injury, regardless of the type of insult. Thus, hepatocytes from zone 2 undergo a clonal expansion to regenerate the liver after chronic CCl4 administration, DDC feeding, PHx and coronavirus infection. Our extensive characterization concurs with the recent findings documented by He et al. and Wei et al. who induced pericentral damages with acute and chronic CCl4 treatments and periportal injuries through the bile duct ligation and DDC feeding models. In addition, beyond hyperplasia, we observed higher hypertrophy in Mid hepatocytes of expanded clonal clusters repopulating the liver architecture in response to damage. We identified significant increased clone and hepatocyte size after chronic CCl4 exposure, especially after 6 weeks of recovery from the cessation of repeated injections. In this model, we observed that Mid hepatocytes contributed to remodeling liver architecture reaching the maximum peak of proliferation at 6 weeks after CCl4 injury. The second population showing a high proliferative rate were PC hepatocytes. Interestingly, proliferation of the PP population is observed to begin 2 days after CCl4 treatment, collaborating with Mid hepatocytes to reshape the organ during the generation of new hepatocytes. However, at day 3 post CCl4 treatment, PP hepatocyte proliferation has decreased while Mid continue to divide, generating new PC hepatocytes. Mid areas along with PC hepatocytes continue to divide at a high rate on 6th day post CCl4 treatment. One week later, at day 12th post CCl4, the liver lobule is fully recovered. These data support a model in which PP hepatocytes begin to expand into the PC region upon repeated damage. Mid hepatocytes also contribute significantly to expanding and reorganizing the liver lobule by generating new hepatocytes closer to the CV. Finally, these new hepatocytes also increase their proliferative activity, locally generating PC hepatocytes until the liver architecture is restored. Thus, combining our acute injury CCl4 model with our chronic injury models, the results support a model where non-damaged hepatocytes in PP areas are the initial responders during liver regeneration, which in combination with Mid hepatocytes, generate large clones that expand radially to PC regions where these new hepatocytes continue to undergo increased in hyperplasia to restore the liver lobule.

Unexpectedly, we observed a significant regression of liver fibrosis during the recovery periods after chronic exposure to CCl4. Pericentral injury has also been associated with activating pericentral hepatic stellate cells, known as the dominant pathogenic cell type responsible for mediating liver fibrosis[91]. We detected remarkable non-hepatocyte CAG-EGFP+ cells at week 14 after prompt cessation of CCl4 treatment which were reduced in number after 2 and 6 weeks of liver regeneration. This process was simultaneously accompanied by the generation of larger hepatocyte clones across the entire liver lobule distance. These data suggest that the overlapping processes of increased hepatocyte growth and cell death of non-hepatocyte cells including pericentral hepatic stellate cells and inflammatory cells may play a key role in fibrosis resolution for liver integrity after CCl4 injury. Further studies are required to identify hepatic cell liver remodeling mechanisms during the reversal of fibrosis and inflammation.

In the model of PHx, we also found the Mid hepatocytes pioneered tissue regrowth through proliferation, similar to the observations found by Chembazi et al.[14]. In addition to hyperplasia of Mid hepatocytes, we also found that liver reconstitution was accompanied by significant hypertrophy of Mid hepatocytes 48 h after PHx, overlapping with the peak of proliferation rate. We also observed hypertrophy of PC hepatocytes during that time point. These findings contradict He et al. who identified streaming of new hepatocytes from zone 1 to 3 during liver regeneration dynamics[8]. Surprisingly, after PHx, we detected groups of hepatocyte-like cells positive for the non-hepatocyte marker CAG-EGFP indicating that these cells derived from non-hepatocyte cells. These unexpected cells increased in number

over liver regeneration dynamics, and they were mostly allocated in the Mid region, coinciding with both, the proliferative hepatocyte peak and the most proliferating area of the liver lobule. Given that the most immediate cell type identified to undergo transdifferentiation mechanism is the cholangiocyte[92], we used the cholangiocyte marker CK19 to potentially detect transdifferentiation from cholangiocyte-like cells to hepatocyte-like cells. However, these cells did not result positive for the cholangiocyte marker CK19. Furthermore, it is unlikely that these cells came from cholangiocytes given that they seem to arise from Mid region. In addition, we identified hepatocyte cells-like positive for both markers, CAG-EGFP non-hepatocyte and Rainbow colored-hepatocyte markers during the regenerative response after PHx. Mechanisms of cell fusion that have been previously defined in the liver[22] between hepatocytes and non-hepatocytes could explain the existence of this phenomenon. Further immunohistochemical studies using specific markers of endothelial cells, bone marrow cells and hepatocytes are required to elucidate the nature of these unexpected cells, as residential endothelial cells[93] and bone marrow cells[94] have been proposed to fuse with hepatocytes in the adult mouse liver. Nevertheless, our study also reports in vivo transdifferentiation and cell fusion mechanisms during full restauration of the hepatic regenerative process after the PHx model. It is noteworthy that we did not observe any mechanism of transdifferentiation between cholangiocytes and hepatocytes after DDC feeding to induce cholestasis and proliferation of biliary epithelial cells as some authors have reported[92,95–97]. Contrary to their findings, our results revealed that the DR caused activation of Mid hepatocytes expanded to regain liver integrity. Our work interrogates clonal expansion, ploidy and hypertrophy of hepatocytes by applying a multicolor lineage tracing strategy after coronavirus infection. We demonstrated that Mid hepatocytes maintain their regenerative capacity through hyperplasic and hypertrophic growth to recover liver integrity after coronavirus infection. In addition, PC hepatocytes also underwent notable hypertrophy to cope liver architecture. Despite we found massive inflammatory infiltrates across the liver lobule, we did not observe any alteration of bile ducts or mechanisms of or transdifferentiation or cell fusion. The elucidation of the reparative mechanistic and hepatocyte dynamics during coronavirus-induced liver damage, may have relevant implications about potential therapeutic interventions based on the past situation with the COVID-19 pandemic, where liver injury has been reported in patients infected with Sars-CoV2[27].

In terms of ploidy changes, we observed increases in the diploid population of new clones generated after chronic CCl4 treatment and coronavirus infection. These diploid clones may have formed by ploidy reversal of polyploid hepatocytes[12] as the hepatocyte polyploid population was reduced in both models. Although diploidy does not significantly affect hepatocyte proliferation in the healthy liver during homeostasis, it might provide a growth advantage in the chronically injured liver[5,15]. Thus, in our models of chronic CCl4 treatment and coronavirus infection, diploid hepatocytes can enter the cell cycle earlier and complete it more rapidly than polyploid hepatocytes. In fact, some studies have found that diploid hepatocytes accelerate liver regeneration in models of liver resection and compensatory regeneration after acute injury[98].

Our study also provided mechanistic insights into interactions between non-hepatocyte cells and hepatocytes to recover liver homeostasis and repair the tissue after acute CCl4 injury. We found through our cell–cell interaction and histological methods that non-hepatocyte cells significantly communicate more with each other after CCl4 damage, and this was accompanied by a peak of proliferation of both non-hepatocytes and hepatocytes. However, although hepatocytes proliferated massively, they reduced their interaction with the rest of the hepatic cells. Thus, while non-hepatocyte cells divide and interact strongly with the rest of the liver cells to attenuate the lesion, undamaged mature Mid hepatocytes exclusively proliferate to restore

the hepatic architecture. In this context, our data would support a model where the hepatocytes act as responder cells of the signals originated from non-hepatocytes cells that are the major drivers of the liver remodeling.

Our analysis of ligand-receptor pairs suggests that the Gal-9-CD44 pathway may play a relevant role in regulating liver remodeling after injury, as Gal-9 and its receptor CD44 resulted upregulated in the hepatic cells after CCl4 treatment compared to control mice. These data suggest that increased Gal-9 either soluble or on the hepatocyte surface can activate CD44 to strengthen the interaction between hepatocytes and other liver cells that have upregulated CD44 such as T-cells, NK cells, neutrophils, monocytes, macrophages and dendritic cells. These non-hepatocyte cells may secrete paracrine signals that can attenuate the immune response and re-adjust the hepatocyte plasticity to increase the proliferation rate of Mid hepatocytes to regenerate the tissue. We highlighted the upregulation of the ligand Gal-9 and its receptor CD44, molecules that have been known previously to regulate several biological functions including liver homeostasis[35,99]. Particularly, galectin proteins are glycan-binding proteins that are known to promote cell–cell adhesion[99], cell migration[99], cell-cycle progression[99], liver injury attenuation[99], T-cell apoptosis[99], hepatocyte apoptosis suppression[99], and liver regeneration[100]. Moreover, Gal-9 regulates a variety of biological functions in immune cells[35] and endothelial cells[101] that are instrumental to the maintenance of hepatic homeostasis[35]. Furthermore, Gal-9-CD44 signaling also mediates migration, invasion, proliferation, survival and metastasis during tumor progression in hepatocellular carcinoma[102]. Thus, Gal-9-CD44 pathway may exert fundamental effects on non-hepatocyte cells for tissue restitution and hepatocytes to proliferate and regenerate the liver after injury. Further studies are required to fully understand the role of Gal-9-CD44 on the trafficking of non-hepatocyte cells and proliferation of Mid hepatocytes during liver regeneration.

In summary, our findings support that Mid hepatocytes have more sensitivity than their counterparts PC or PP to detect and respond to cell turnover requirements and tissue impairments in liver damage, as previously demonstrated[8,13]. However, when liver is injured, Mid hepatocytes received help from PC or PP hepatocytes to repair the lesion. Our manuscript goes beyond recent previous studies to demonstrate that in addition to hyperplasia of Mid hepatocytes, liver reconstitution is also accompanied by hypertrophy of these hepatocytes after chemical, physical and coronaviral damages. In addition, the polyploid hepatocyte population is enriched in the healthy rather than repaired liver, where reductions in hepatocyte ploidy seem to participate in liver architecture dynamics during its regeneration, after severe chemical and viral injuries. The increased proliferative capacity of mature Mid hepatocytes demonstrated here and in recent studies[8,13,25] rules out the fact of a stem cell-like or non-hepatocyte cell-type involved in restoring hepatic histology. Moreover, we propose that Gal-9 expressed in Mid hepatocytes and Gal-9-CD44 pathway expressed in other liver cells are the drivers of hepatocyte proliferation and liver reconstitution activating autocrine and paracrine signals. Finally, our findings of transdifferentiation or cell fusion activated during PHx provide provocative insights on the current field of liver biology about other key mechanisms participating in liver regeneration that would need further consideration. The data presented herein of liver heterogeneity in response to hepatic injuries provide significant advances in our understanding of liver cell-based regenerative therapies for patients diagnosed with liver disorders, given the growing incidence of cirrhosis, viral hepatitis, fatty liver diseases, and liver cancer. Ongoing studies are warranted to better characterize the Gal-9/Gal-9-CD44 signaling activities in Mid hepatocytes and other liver cell types to develop alternative therapeutic interventions using glycan-binding proteins.

## Methods

### Mice

Experiments were conducted under the ethical guidelines and protocols approved by IACUC (Institutional Animal Care and Usage Committee) in Yale University School of Medicine (Animal protocol, 2022-11577). Homozygous Rosa 26-Rainbow Cre-mediated recombination mice (*Rosa26$^{rbw}$*)[17] were obtained from Prof. Daniel Greif at Yale School of Medicine and previously generated by Prof. Irv Weissman at Stanford University. The Rainbow construct of the *Rosa26$^{rbw}$* mice is knocked-in into Rosa26 locus and consists of 3 loxP variants: lox2271, loxN, and loxp, and subsequent cDNA for the Enhanced Green Fluorescent protein (EGFP) followed by cDNAs for mCerulean, mOrange and mCherry, which are proceeded by lox2271, loxN, and loxP, respectively. In the absence of Cre-recombinase-mediated recombination, CAG-EGFP is expressed constitutively resulting in each liver cell permanently expressing green color (Fig. 1A). *Rosa26$^{rbw}$* mice were crossed with homozygous TMX-inducible albumin Cre mice (*Alb-CreERT2*)[18] provided by Prof. Daniel Metzger to generate heterozygous *Alb-CreERT2 Rosa26$^{rbw}$* mice, which specifically showed the Rainbow colors in the cells that expressed *albumin* (hepatocytes). TMX administration resulted in hepatocyte-specific recombination, being these cells permanently labeled and randomly recombined to express 1 of 3 colors; yellow (mOrange), red (mCherry) and/or light blue (mCerulean). As Cre recombinase expression was restricted to hepatocytes, non-hepatocyte cells including cholangiocytes, stellate cells, immune and endothelial cells remained labeled in green (CAG-EGFP). *Alb-CreERT2 Rosa26$^{rbw}$* mice were subjected of the liver damage models used in our study and they were euthanized by isoflurane inhalation. Agarose gel electrophoresis was performed to confirm the genotype of *Rosa26$^{rbw}$*, and *Alb-CreERT2* mice (See Supplementary Fig. 2). All mice were housed in a barrier animal facility with a constant temperature and humidity in a 12-h dark/light cycle. All mice were fed with a standard chow diet [CD (Envigo 2018S)] and water and food were provided ad libitum.

### Genotyping

Ear biopsies were removed from weaned mice and DNA samples were heat at 95 °C in 50 mM NaOH buffer for 30 min followed by 1 M Tris HCl (pH 6.8) addition to neutralize tissue digest. For *Rosa26$^{rbw}$* PCR, DNA was amplified using 35 cycles (94 °C: 30 s, 61 °C: 60 s, 72 °C: 60 s) with the following primers; 1: 5′CTCTGCTGCCTCCTGGCTTCT3′, 2: 5′CGAGGCGGATCACAAGCAATA3′ and 3: 5′TCAATGGGCGGGGGTCGTT3′. Products were fractionated by gel electrophoresis (150 V, 40 min) using 2% agarose in bionic buffer, with predicted amplicons of 330 base pairs for wild-type mice and 250 base pairs for *Rosa26$^{rbw}$* mice (Fig. S2A). For Alb-CreERT2 PCR, DNA was amplified using 39 cycles (94 °C: 30 s, 51.7 °C: 30 s, 72 °C: 30 s) with the following primers; 1: 5′GCGGTCTGGCAGTAAAAACTATC3′, 2: 5′GTGAAACAGCTTGCTGTCACTT3′, 3: 5′GTAGGCCACAGAATTGAAAGAT3′ and 4: 5′GTAGGTGGAAATTCTAGCATCA3′. Products were fractionated by gel electrophoresis (150 V, 40 min) using 2% agarose in bionic buffer, with predicted amplicons of 229 base pairs for wild-type mice and 444 base pairs for *Alb-CreERT2* mice (Supplementary Fig. 2B).

### Immunofluorescence

Following euthanasia, livers were fixed in 10% formalin (Thermo Fisher Scientific) in Sodium Chloride (NaCl), 0.9% (w/v) Aqueous, Isotonic Saline (RICCA Chemical) overnight at 4 °C. Tissue was then incubated in 15 and 30% sucrose gradients overnight each diluted in 0.9% (w/v) NaCl, embedded in optical cutting temperature compound (Tissue Tek), and stored at −80 °C. Liver lobules were cryosectioned (6 μm) in the transverse axis, and sections were washed with 0.1% Triton X-100 in phosphate-buffered saline (PBS) solution (PBS-T) and immersed in mounting medium with the nuclear fluorescent dye 4′,6-Diamidino-2-

phenylindole dihydrochloride (DAPI) (Vector laboratories). Slides were immediately visualized using fluorescent filters for DAPI and the Rainbow colors (mCerulean, mOrange, mCherry and CAG-EGFP). Images were acquired with the Nikon microscope (Eclipse 80i upright fluorescent or Eclipse TS100 inverted). For image processing, analysis and cell counting, Adobe Photoshop and Image J (Fiji) software were used.

## Immunohistochemistry

Similar to the IF method, livers were fixed in 10% formalin (Thermo Fisher Scientific) in 0.9% (w/v) NaCl (RICCA Chemical) overnight at 4 °C. Tissue was then incubated in 15 and 30% sucrose gradients overnight each diluted in 0.9% (w/v) NaCl, embedded in optical cutting temperature compound (Tissue Tek), and stored at −80 °C. Liver lobules were cryosectioned (6 μm) in the transverse axis and sections were washed twice with PBS during 3 min, incubated with 1% hydrogen peroxidase in PBS for 30 min and blocking with normal goat or donkey serum in PBS for 30 min. After that, liver sections were incubated overnight at 4 °C with the following primary antibodies: Anti-Cytokeratin 19 (Abcam, no. 133496; 1:100), anti-Glutamine Synthetase (Abcam, no. 197024; 1:50), anti-CD31 (Abcam, no. 28364; 1:100), anti-CD68 (Abcam, no. 125047; 1:200), anti-Ki67 (Abcam, no. 15580; 1:100) and anti-CD45 (Novus Biologicals no. AF114; 1:100). The following day, sections were washed with 3 times with PBS for 5 min and incubated 30 min at room temperature with the biotinylated secondary antibodies (Rockland, anti-rabbit no. 611-106-122 and anti-goat no. 605-4613; 1:500 in PBS). Then, slides were washed 3 times with PBS for 2 min and incubated at room temperature with streptavidin peroxidase conjugated (Rockland, no. S000-03) for 30 min at 1:500 in PBS. Finally, sections were incubated with 3,3′-Diaminobenzidine (DAB) Substrate (Rockland, no. DAB-10) for 3–8 min and counterstained with hematoxylin (Millipore Sigma). For apoptosis staining, we used the TUNEL Assay Kit- HRP-DAB (Abcam, no. 206386). Consecutive sections were stained with H&E and Sirius red through the Yale Pathology Tissue Services core to evaluate liver morphology, necrosis and fibrosis. Images were taken with an EVOS microscope. Quantification of stained areas was performed with ImageJ (Fiji) software.

## Liver damage models

For all liver injury models, 8-week-old *Alb-CreERT2 Rosa26^rbw* mice were injected with 3 injections of TMX to activate Rainbow fluorophore expression. After a washout period of 2 weeks, mice were subjected to CCl4, DDC diet, PHx or MHV-A59. _Acute CCl4:_ an-IP injection of CCl4 (Sigma-Aldrich no. 289116; NH$_2$COCH$_2$) was administered at a dose of 1 μL/g body weight to 10-week-old male *Alb-CreERT2 Rosa26^rbw* mice. CCl4 was diluted in corn oil at a final volume of 50 μL. Untreated mice received a dose of 50 μL of corn oil. After 2, 3, 6 and 12 days, mice were euthanized and livers were collected for histological analysis. 2-4 mice per time point were used for CCl4 administration or control untreated groups. _Chronic CCl4_: CCl4 (Sigma-Aldrich no. 289116; NH$_2$COCH$_2$) was administered by IP injection at a dose of 0.25 μL/kg every 3 days for a total of 4 weeks, at 10-week-old female *Alb-CreERT2 Rosa26^rbw* mice. CCl4 was diluted in corn oil at a final volume of 50 μL. Untreated mice received weekly doses of 50 μL of corn oil. At week 14, 16 and 20 of age, mice were euthanized, and livers were collected for histological analysis. 4–5 mice per time point were used for CCl4 administration or control untreated groups. _PHx_: 10-week-old female *Alb-CreERT2 Rosa26^rbw* mice were subjected to 2/3 PHx by the Yale Liver Center Facility at Yale School of Medicine. At 0, 24, 48, 72 and 162 h after surgery, mice were euthanized, and livers were collected for histological analysis. 2 mice were used per time point. _DDC feeding_: 10-week-old female *Alb-CreERT2 Rosa26^rbw* mice were fed with a 0.1% DDC-supplemented diet (Sigma-Aldrich, no. 137030) or a chow diet for 6 weeks (*N* = 4 mice per group). After this period, mice were

euthanized, and livers were collected for histological analysis. _MHV-A59 infection_: 10-week-old male *Alb-CreERT2 Rosa26^rbw* mice were intranasally inoculated with MHV-A59 (10^6 PFU). After 1- or 4-weeks mice were euthanized and livers were collected for histological analysis (*N* = 3 mice per group). For all the injured model generated, TMX was intraperitoneally injected to 8-week-old *Alb-CreERT2 Rosa26^rbw* mice as a 20 mg/mL solution in corn oil at a dose of 100 μL per mouse during 3 consecutive days.

## Clonal quantification analysis

To generate the graphs in Figs. 1G, H, 2D, E, 3F, H, S3F, 4H, 5D, E, S5A, 6F, G and Supplementary Fig. 7A, B, 4–5 photos containing 3–4 areas per field of view of 10× corresponding to a liver lobule (the distance comprised between a periportal and a pericentral vein) were analyzed per mouse, for a total of 3–4 mice per group within a same time point (except for the PHx, MHV-A59 models and Supplementary Fig. 7 that it was used 2 mice). We named hepatocyte clone a hepatocyte formed by only one hepatocyte (individual cell) or more than one hepatocyte colored with the same color (mCherry, mCerulean or mOrange). We counted the number of all clones and their size (number of cells per clone) found in the PP, Mid and PC regions using the tool polygon selection with the Image J Fiji software. We kept the same size of the polygon selection generated with Image J Fiji software for all the images analyzed. Clones containing a cell located within 4 cell distances or 4 layers from the portal vein were classified as periportal; clones containing a cell within 3 cell distances or 3 layers from the central vein were classified as pericentral; clones not meeting either criterion were classified as Mid. We considered PC clones those containing a cell within 3 cell distances from the CV based on our GS staining, as we also saw expression of this pericentral enzyme by hepatocytes located in the third layer from the CV distance. Number of cells per clone was determined based on clone morphology and nuclear staining with DAPI. We distinguished clones containing mononucleated cells from multinucleated individual-cell clones based on whether the cell shape appeared to be separated cells with a nucleus each one or whether the nuclei were sharing the same cytoplasm within a single cell. CAG-GFAP staining helped us to outline the plasma membrane. Data were tabulated and analyzed in Microsoft Excel and GraphPad Prism.

## Colored hepatocytes in % per fluorophore (mOrange, mCherry and mCerulean) and per area of the liver lobule (PP, Mid and PC)

To generate this graph (Fig. 1F), we counted and calculated the sum of the number of PP, Mid and PC hepatocytes per Rainbow color, and we divided by the total hepatocyte number per Rainbow color. We then multiply by 100. Data were tabulated and analyzed in Microsoft Excel and GraphPad Prism.

## Hepatocyte ploidy analysis

To generate the graphs in Figs. 1J, 2L, 3P, S3H, 4N, 5K, S5G and 6K), 5 areas per field of view of 10× corresponding to a liver lobule (the distance comprised between a periportal and a pericentral vein) were analyzed per mouse, for a total of 3 mice per group within a same time point (except for the PHx and MHV-A59 models where it was used 2 mice). To define the hepatocyte ploidy, hepatocytes expressing only one Rainbow color (mCherry, mCerulean or mOrange) were considered diploid. Meanwhile, hepatocytes expressing more than one color were considered polyploid. Within these 2 groups, we also quantified mononucleated *vs.* multinucleated cells based on DAPI staining. Supplementary Fig. 8 shows representative detailed images of diploid and polyploid mono and multinucleated hepatocytes with marked cell boundaries. We analyzed all hepatocytes in PP, Mid and PC regions following the criterion defined in the previous section. Data were tabulated and analyzed in Microsoft Excel and GraphPad Prism.

## Single cell-RNA-sequencing analysis

Preparation of single-cell suspension: Hepatic single-cell suspensions were prepared for submission for 10X single-cell RNA-sequencing and flow cytometry. Livers were perfused firstly with digestion cocktail containing 1.5 mg/mL Collagenase I (Sigma; SCR103) and 0.5 mg/mL DNase I (Roche; 4716728001) from the portal vein until tissue softens and then perfused with DMEM medium containing 10% FBS. Tissues were removed and placed in cold media before shaking to dissociate hepatic single-cell suspensions. Isolated suspensions were centrifuged at 60 G for 2 min with 2 brakes to separate parenchymal fraction (Hepatocytes, Cholangiocytes) and non-parenchymal (NPC) fractions (Endothelial, Immune cells etc.). For parenchymal fractions, a 35% Percoll (Sigma; P1644) gradient was used to enrich for live cells after spinning at $500 \times g$ for 5 min. NPC fractions can be directly processed for flow cytometry or further enriched for single-cell RNA sequencing. To enrich for live NPCs, cells were stained with LIVE/DEAD Fixable Aqua Dead Cell Stain Kit (Invitrogen; L34957) before sorting on the FACSAria II Cell sorter. Purified parenchymal and NPC fractions were mixed at a 1:1 ratio before submission for single-cell sequencing. Droplet-Based Single-Cell RNA Sequencing Library Construction: Live cell enriched Parenchymal and NPC cells were encapsulated into droplets and processed following the manufacturer's specifications using 10X Genomics GemCode technology. Equal numbers of cells per sample were loaded on a 10x Genomics Chromium controller instrument to generate single-cell Gel Beads in emulsion (GEMs) at the Yale Center for Genome Analysis. Lysis and barcoded reverse transcription of polyadenylated mRNA from single cells were performed inside each GEM followed by complementary DNA (cDNA) generation using the Single-Cell 3' Reagent Kits version 2 (10X Genomics). Libraries were sequenced on an Illumina HiSeq 4000 as $2 \times 100$ paired-end reads. Pre-processing of single-cell RNA-seq data: Single-cell RNA data from this project were processed using CellRanger software (version 2.1.1) as previously described[103]. Firstly, sample demultiplexing, aligning read to the mouse genome (University of California Santa Cruz mm10 reference genome) with Software Tools for Academics and Researchers (STAR) and unique molecular identifier (UMI) processing was performed. The raw gene expression matrix was filtered with the following criteria. Cells with over 20% mitochondrial gene expression in UMI counts were removed; cells with under 300 detected genes were removed; cells with more than 25,000 UMI were removed. After filtering, a total of 7106 cells from control treatment and 4747 cells from CCL4 treated samples were identified for further analysis. Dimension reduction, unsupervised clustering, and cell cluster annotation: Processed gene expression matrix with all retained cells for each sample was imported to the Seurat R package (v3.1.0) for downstream analyses[104]. Data were normalized using the 'NormalizeData' function, in which UMI counts for each gene were divided by the total UMI counts in each cell and multiplied by the scale factor of 1000, following by natural-log transformation with adding a pseudo count of 1 for each gene. Based on the normalized expression matrix, 2000 most variable genes were identified using the 'FindVariableFeatures' function with the 'vst' method. High variable genes were applied for the principal component analysis (PCA) to identify the top 50 principal components using the 'RunPCA' function of Seurat, which was then applied to dimension reduction using the 'RunUMAP' function in Seurat. Uniform Manifold Approximation and Projection (UMAP) visualization indicated cells from different samples were well mixed into the shared space[105]. For clustering, we applied built a Shared Nearest Neighbor (SNN) graph using Principal components (PCs) 1 to 50 and $k = 25$ nearest neighbors, and then the Louvain clustering algorithm was used to group the cells into different clusters. Cell clusters were annotated based on top differentially expressed marker genes and mapped to established cell signatures. The scRNA-seq data has been publicly deposited to GEO (GSE250568).

## Ligand-receptor cell communication analysis

To identify potential intercellular interactions between different cell types, we utilized the R package CellChat version 1.6.0[106]. In brief, CellChat object was created by feeding the processed Seurat object into the createcellchat function and processed using its standard pipeline. CellChatDB.mouse was loaded and differentially expressed genes and interactions were identified in the CellChat object via IdentifyOverExpressedGenes and IdentfyOverExpressed interactions[106], respectively. The CellChat algorithm was then run to calculate the communication probability and infer cellular communication network via ComputeCommunProb and ComputeCommunProbPathway[106].

## Statistical analysis

All data were analyzed by unpaired Student's t-test comparisons or one-way ANOVA with repeated measures followed by post hoc tests, as appropriate, using GraphPad Prism 8.0 (GraphPad Software, Inc.). All numerical values are presented as mean ± SEM and $p < 0.05$ was considered statistically significant.

## Reporting summary

Further information on research design is available in the Nature Portfolio Reporting Summary linked to this article.

## Data availability

The scRNA-seq data generated in this study have been deposited in the Gene Expression Omnibus GEO database under accession code GSE250568. All other data that support the findings of this study are within the article and its Supplementary Information and Source Data files. Source data are provided with this paper.

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

## Acknowledgements

We thank the Yale Liver Center Facility at Yale School of Medicine for conducting the partial hepatectomy model in mice. We thank Prof. Daniel Metzger for providing with the *Alb-CreERT2* mice and Prof. Susan Compton for supplying with the MHV-A59. This work was supported by grants from the NIH (R35HL135820 to C.F.H.).

## Author contributions

I.R.M. designed and performed the experiments, analyzed and interpreted the data, prepared figures and wrote the manuscript. J.G. designed and generated the mouse models of liver injuries. H.Z. performed the cell–cell communication analysis. J.S. performed the mouse infection with MHV-A59 and processed the scRNA-seq data. A.B. helped to breed mice and maintain mouse colonies. D.M.G. provided Rosa26rbw mice and advised to interpret the multicolor lineage tracing experiments. I.K. provided technical support with the fluorescence microscope used to visualize the rainbow fluorophores. R.G.K. interpreted the data and reviewed the manuscript. Y.S. supported and advised the design and interpreted the data. C.F.H. conceived the project, interpreted the data, reviewed the manuscript and acquired funding for the project. All the authors revised the manuscript and agreed to the published version of the manuscript.

## Competing interests

The authors declare no competing interests.
