## [Peer Review File · Nature Communications]

Heterogeneity of hepatocyte dynamics restores liver architecture after chemical, physical or viral damageREVIEWER COMMENTS

Reviewer #1 (Remarks to the Author):

This article uses a similar experimental paradigm as (Chen et al., 2020). The use of a stable knock-in Alb-CreERT2 mouse line elevates the labeling efficiency of the Rainbow allele and results in beautiful imaging data. The language of the article is great with a few minor mistakes in figure legends and the Method part. The amount of workload is also embodied by the massive quantitative data that the authors extracted from the images, probably in an attempt to draw a line to the debate on the heterogeneity of hepatocyte proliferation and regeneration in terms of spatial distribution, ploidy, models and cells of origin.

1. The first question is the clonal quantification. Since the labeling efficiency of hepatocytes by different color is very high, the images seem to be very difficult to tell one clone from another in some regions, as clones in similar color are located close to each other. Dilution or more sparse labeling might be appropriate to tell the true clones.

2. In the Clonal quantification analysis section of the Methods part, the authors state “clones containing a cell within 3 cell distances or 3 layers from the central vein were classified as pericentral”, while (Chen et al., 2020) stated “clones containing a cell within 2 cell distances from the central vein were classified as pericentral” in their Methods. The reviewer is interested in the reason that the authors expand the range of pericentral hepatocytes.

Moreover, the blue-arrowhead-indicating Ki67+ PC hepatocyte nucleus in the right panel of Fig. 1I seems to be midzonal to the reviewer, as it is already 3-4 layers of nuclei (difficult to tell whether it is hepatocyte or non-hepatocyte nuclei) from the CV. And as the GS IHC is shown in Fig.1B, the 2-cell distance seems more reasonable to the reviewer. Due to the particularity of Rainbow, it is not feasible to perform IF of multiple liver zone markers, but it at least shows the potential subjectivity of the quantitative analysis and disparity of PC hepatocyte definition between the authors and other research teams. The reviewer would like the authors to provide multiple representative images with cell-counting marks and clearly marked zonation boundaries.

3. The reviewer would also like the authors to show representative images of multi-/mono-nucleation in polyploid hepatocytes under different conditions and marked cell boundaries for midzonal hepatocyte hypertrophy after injury because the quantifying methods seem quite subjective.

4. In Fig.3D, the authors added arrows indicating hepatocyte clonal expansion routes after CCl4 chronic injury and stated "after chronic CCl4 exposure...we observed that PC hepatocytes along with Mid also contributed to remodeling liver architecture". Given that the authors might have overestimated PC hepatocyte proliferation, the reviewer questions the biological meaning of PC hepatocytes expanding all the way toward the PP region when PC hepatocytes are repeatedly damaged by CCl4. To the reviewer, the expansion might be mostly starting from PP and midzonal regions, while the reviewer does not deny the regenerative capacity of PC hepatocytes under this condition.

References

Chen, F. et al. (2020) 'Broad Distribution of Hepatocyte Proliferation in Liver Homeostasis and Regeneration', *Cell Stem Cell*, 26(1), pp. 27-33.e4. Available at:

<https://doi.org/10.1016/j.stem.2019.11.001>.

He, L. et al. (2021) 'Proliferation tracing reveals regional hepatocyte generation in liver homeostasis and repair', *Science*, 371(6532). Available at:

<https://doi.org/10.1126/science.abc4346>.

Raven, A. et al. (2017) 'Cholangiocytes act as facultative liver stem cells during impaired hepatocyte regeneration', *Nature*, 547(7663), pp. 350–354. Available at:

<https://doi.org/10.1038/nature23015>.

Reviewer #2 (Remarks to the Author):

In this study, Ruz-Maldonado et al investigated the heterogeneity of hepatocytes dynamics during liver homeostasis and repair. They used Alb-CreER;R26-Rainbow mouse model to

perform clonal analysis of hepatocyte proliferation, and show that hepatocytes located in midlobular zone contribute to liver homeostasis and regeneration. The authors demonstrated that hepatocytes undergo hypertrophy in response to injury. Furthermore, the authors show that non-hepatocytes fuse with hepatocytes after PHx. However, it has been reported that midlobular zone is the main source of new hepatocytes during liver homeostasis and regeneration (PMID: 33632817, PMID: 33632818). Therefore, this study lacks conceptual novelty and needs to present more new biological insights into the hepatocyte proliferation on liver regeneration.

Major comments:

1. The authors used Alb-CreER;R26-Rainbow mouse model in many analyses, and almost all of hepatocytes were labeled by 3 fluorescent proteins (mCherry, mCerulean and mOrange). It is hard to distinguish individual hepatocyte and clones consist of 2 or more hepatocytes from the figures. Strictly, this mouse model is not good for analysis of clonal expansion of hepatocytes.
2. The authors detected hepatocyte ploidy in many analyses, but the authors did not show any biological insights of hepatocyte ploidy into the hepatocyte proliferation on liver regeneration. Furthermore, the authors considered hepatocytes that expressed only one Rainbow color to be diploid. This is not rigorous. Detection of hepatocyte ploidy should be performed based on Hoechst intensity (DOI:10.1038/nature09414).
3. Did the authors detect EGFP+ hepatocytes at 0h after PHx? There appear to be EGFP+ hepatocyte in FigS5A. The efficiency of tamoxifen-mediated recombination of Cre-loxP is unlikely to reach 100%. There may be individual EGFP+ hepatocytes before PHx. And hepatocytes could be multinuclear, so some of EGFP+ hepatocytes could be multicolor. PHx induces hepatocytes proliferation. Individual EGFP+ hepatocytes could proliferate to form clones consist of 2 to 6 cells after PHx. Therefore, cell fusion might not be the origin of these EGFP+ hepatocytes during PHx. The author should perform additional experiments to detect cell fusion during PHx and quantify the percentage of hepatocytes formed by cell fusion. Moreover, it has been reported that bone marrow cells and endothelial cells could contribute to hepatocytes through cell fusion (PMID: 12665832 14555960, 32487457). Which type of non-hepatocytes fuse with hepatocytes in this study?
4. The authors state that "Taken together, these results confirm that there is not any

mechanism of transdifferentiation between biliary and hepatic epithelia as we hypothetically could have found in our multicolor lineage tracing Alb-CreERT2 Rosa26^{rbw} mice (Figures S4A-B).” It has been reported that hepatocytes could be converted into biliary-like cells in cholestatic liver injury (PMID: 32971004). The authors may miss this transdifferentiation of hepatocytes into BECs because of the limitation of technology. The authors should perform co-immunostaining for fluorescent proteins and CK19 (CK19 could be set at 647 channel).

Minor comments:

1. Could the authors clarify statistical methods of Fig1F.
2. Does the statistical picture in Fig5H represent the sum of Ki67⁺ hepatocytes at 24h, 48h and 72h? The author should show the number of Ki67⁺ hepatocytes at 24h, 48h and 72h separately.

REVIEWER COMMENTS

We thank the reviewers for their careful analysis of our study and their insightful comments. We are pleased by the overall enthusiasm for this work and are confident that we were able to address all the referees' major concerns. In the revised manuscript, we performed new experiments as both reviewers suggested to clarify the interpretation of our results, and we also included a new scRNA-seq analysis and a cell-cell interaction method derived from the scRNA-seq dataset to gain additional novelty and further mechanistic insights into the hepatocyte proliferation on liver regeneration. Through these new methods, we have found that hepatocytes upregulate Galectin-9 (Gal-9) to tightly interact with non-hepatocytes cells that also overexpress Gal-9 and its receptor CD44 for driving immune homeostasis and tissue remodeling after liver injury. We updated a more detailed discussion of how the novel findings we have presented impact the interpretation of previous outcomes and the overall progression of the field. Below is a point-by-point response detailing how we modified the manuscript and the experimental approach used in response to the comments made by the reviewers.

Reviewer #1 (Remarks to the Author):

This article uses a similar experimental paradigm as (Chen et al., 2020). The use of a stable knock-in Alb-CreERT2 mouse line elevates the labeling efficiency of the Rainbow allele and results in beautiful imaging data. The language of the article is great with a few minor mistakes in figure legends and the Method part. **The amount of workload is also embodied by the massive quantitative data that the authors extracted from the images, probably in an attempt to draw a line to the debate on the heterogeneity of hepatocyte proliferation and regeneration in terms of spatial distribution, ploidy, models and cells of origin.**

1. The first question is the clonal quantification. Since the labeling efficiency of hepatocytes by different color is very high, the images seem to be very difficult to tell one clone from another in some regions, as clones in similar color are located close to each other. Dilution or sparser labeling might be appropriate to tell the true clones.

To address this point, we used diluted TMX concentrations, 0.1 mg/mL and 0.5 mg/mL, based on our dose-response test for Cre recombinase activation indicated in **Supplementary Fig.1**. We then reproduced the acute CCl₄ damage in our mice and we histologically analyzed the liver sections after 3 and 6 days of CCl₄ treatment to clearly visualize the formation of the clones. We included this data as a new Supplementary Figure (**Supplementary Figure 7**) showing detailed image views of the clones and we incorporated this information into the main text.

2. In the Clonal quantification analysis section of the Methods part, the authors state “clones containing a cell within 3 cell distances or 3 layers from the central vein were classified as pericentral”, while (Chen et al., 2020) stated “clones containing a cell within 2 cell distances from the central vein were classified as pericentral” in their Methods. The reviewer is interested in the reason that the authors expand the range of pericentral hepatocytes. Moreover, the blue-arrowhead-indicating Ki67+ PC hepatocyte nucleus in the right panel of Fig. 1I seems to be midzonal to the reviewer, as it is already 3-4 layers of nuclei (difficult to tell whether it is hepatocyte or non-hepatocyte nuclei) from the CV. And as the GS IHC is shown in Fig.1B, the 2-cell distance seems more reasonable to the reviewer. Due to the particularity of Rainbow, it is not feasible to perform IF of multiple liver zone markers, but it at least shows the potential subjectivity of the quantitative analysis and disparity of PC hepatocyte definition between the authors and other

research teams. The reviewer would like the authors to provide multiple representative images with cell-counting marks and clearly marked zonation boundaries.

As the reviewer points out, we have considered PC clones those containing a cell within 3 cell distances from the CV based on our GS staining, as we see expression of this pericentral enzyme by hepatocytes located in the third layer from the CV distance. We agree with the reviewer that GS expression is more significant and consistent in layers 1 and 2 than in layer 3 of the CV, but many hepatocytes express detectable levels of GS in layer 3, as shown by the control images in **Figures 4F and 2B**. Here we provide 3 more representative images of GS staining (A-C) indicating cell count marks and clearly marked zonation boundaries. Black arrows indicate PC hepatocytes. We have now incorporated this information into the main text on method details to provide greater clarity on our rationale.

3. The reviewer would also like the authors to show representative images of multi-/mono-nucleation in polyploid hepatocytes under different conditions and marked cell boundaries for midzonal hepatocyte hypertrophy after injury because the quantifying methods seem quite subjective.

We thank the reviewer for bringing up this point as this will help readers to better understand our analysis and clearly interpret our findings. We added representative detailed images of diploid and polyploid mono and multinucleated hepatocytes as a new Supplementary Figure (**Supplementary Figure 8**) with marked cell boundaries. We also added detailed images of hepatocyte hypertrophy after 20 weeks of CCl₄ treatment compared to control mice in **Supplementary Figure 9**.

4. In Fig.3D, the authors added arrows indicating hepatocyte clonal expansion routes after CCl₄ chronic injury and stated, "after chronic CCl₄ exposure...we observed that PC hepatocytes along with Mid also contributed to remodeling liver architecture". Given that the authors might have overestimated PC hepatocyte proliferation, the reviewer questions the biological meaning of PC hepatocytes expanding all the way toward the PP region when PC hepatocytes are repeatedly damaged by CCl₄. To the reviewer, the expansion might be mostly starting from PP and midzonal regions, while the reviewer does not deny the regenerative capacity of PC hepatocytes under this condition.

We would like to thank the reviewer for raising this interesting point. To more accurately assess hepatocyte dynamics during liver repair, we added two additional time points after acute CCl₄ damage. We repeated this model using additional mice and we analyzed liver histology after 3 and 6 days of acute CCl₄ treatment compared to vehicle-treated control mice. We then quantified clonal expansion of the liver lobule at 3 and 6 days after CCl₄ and we analyzed hepatocyte morphology and proliferation by H&E and Ki67 staining. We combined these new data in a new Figure (**Supplementary Figure 7**). We concluded that our results support a model in which undamaged hepatocytes from PP zones are the initial responders during liver regeneration, which in combination with Mid hepatocytes, generate large clones that expand radially to PC regions where these new hepatocytes continue to undergo increased hyperplasia to restore the hepatic lobule. In the revised manuscript, we modified the main text in the Results of acute CCl₄

treatment, made a new Figure (**Supplementary Figure 7**) with new time points after CCl4 and hepatocyte analysis, updated the methods, addressed the model of PP and Mid hepatocytes expanding to the PC region when this area is repeatedly damaged by CCl4 in the Discussion. We also changed the directions of the arrows in Figure 3D.

Reviewer #2 (Remarks to the Author):

In this study, Ruz-Maldonado et al investigated the heterogeneity of hepatocytes dynamics during liver homeostasis and repair. They used Alb-CreER;R26-Rainbow mouse model to perform clonal analysis of hepatocyte proliferation, and show that hepatocytes located in midlobular zone contribute to liver homeostasis and regeneration. The authors demonstrated that hepatocytes undergo hypertrophy in response to injury. Furthermore, the authors show that non-hepatocytes fuse with hepatocytes after PHx. However, It has been reported that midlobular zone is the main source of new hepatocytes during liver homeostasis and regeneration (PMID: 33632817, PMID: 33632818). Therefore, this study lacks conceptual novelty and needs to present more new biological insights into the hepatocyte proliferation on liver regeneration.

We thank the reviewer for their suggestions to improve the quality of our manuscript. To gain further conceptual novelty and new biological insights into hepatocyte proliferation during liver regeneration, we generated another model of acute CCl4 liver injury using additional mice (N=4 per condition). We processed samples for histology (N=2 per condition) and performed scRNA-seq and cell-cell interaction analysis (N=2 per condition) at day 6 after acute CCl4 treatment, when peak hepatocyte proliferation is expected to be reached based on our previous results observed in the acute CCl4 model. We found that Gal-9 expressed in Mid hepatocytes and Gal-9-CD44 pathway expressed in other liver cells are the drivers of hepatocyte proliferation and liver reconstitution activating autocrine and paracrine signals. We have included these findings in 4 additional Figures (**Figure 7** and **Supplementary Figures 10-12**). We think that these new results provide novel mechanistic insights of how the liver regenerates in response to injury. We also think that our study contains numerous models of acute and chronic damage to assess hepatocyte clonal expansion, some of them never studied (e.g. coronavirus induce liver damage).

Major comments:

1. The authors used Alb-CreER;R26-Rainbow mouse model in many analyses, and almost all of hepatocytes were labeled by 3 fluorescent proteins (mCherry, mCerulean and mOrange). It is hard to distinguish individual hepatocyte and clones consist of 2 or more hepatocytes from the figures. Strictly, this mouse model is not good for analysis of clonal expansion of hepatocytes.

To address this point, we repeated our acute CCl4 model in additional mice using lower TMX concentrations (0.1 and 0.5 mg/mL) to observe independent clones with increased numbers with single colors. We histologically analyzed the liver sections after 3 and 6 days of CCl4 treatment to clearly visualize the formation of the clones. We included this data as a new Supplementary Figure (**Supplementary Figure 7**) showing detailed image views of the clones or individual hepatocytes and we incorporated this information into the main text.

2. The authors detected hepatocyte ploidy in many analyses, but the authors did not show any biological insights of hepatocyte ploidy into the hepatocyte proliferation on liver regeneration. Furthermore, the authors considered hepatocytes that expressed only one Rainbow color to be

diploid. This is not rigorous. Detection of hepatocyte ploidy should be performed based on Hoechst intensity (DOI:10.1038/nature09414).

Several studies have demonstrated that heterozygous multicolor reporter mice, as our *Alb-Cre Rainbow* mice, are a good model to examine hepatocyte ploidy, including Chen et al., 2020 and Matsumoto et al., 2020, which are references 5 and 19, respectively. We performed our analyses of ploidy here using these previously established techniques. In addition to our original introduction of ploidy analysis in Figure 1C, we added a new Figure (**Supplementary Figure 8**) with representative detailed images of diploid and polyploid hepatocytes to clarify our ploidy analysis. In addition, we have now extended the Discussion to further address the potential biological implication of the ploidy alteration during liver repair in our models of chronic CCl₄ treatment and coronavirus infection.

Regarding the detection of the DNA staining content to identify mononucleated vs multinucleated hepatocytes, we employed DAPI. We chose this over Hoechst as our samples require fixation and DAPI is preferred for fixed cell staining while Hoechst dye is recommended for live cell staining. Regardless, we don't find that hepatocyte nucleus labeling with DAPI or another DNA dye can rigorously determine ploidy status in tissue sections, as all nuclei for a cell may not be included in the section. For this reason, we preferentially measured hepatocyte ploidy depending on the expression of 1 or more than 1 color as previously established for our genetic model system.

3. Did the authors detect EGFP+ hepatocytes at 0h after PHx? There appear to be EGFP+ hepatocyte in FigS5A. The efficiency of tamoxifen-mediated recombination of Cre-loxP is unlikely to reach 100%. There may be individual EGFP+ hepatocytes before PHx. And hepatocytes could be multinuclear, so some of EGFP+ hepatocytes could be multicolor. PHx induces hepatocytes proliferation. Individual EGFP+ hepatocytes could proliferate to form clones consist of 2 to 6 cells after PHx. Therefore, cell fusion might not be the origin of these EGFP+ hepatocytes during PHx. The author should perform additional experiments to detect cell fusion during PHx and quantify the percentage of hepatocytes formed by cell fusion. Moreover, it has been reported that **bone marrow cells** and **endothelial cells** could contribute to hepatocytes through cell fusion (PMID: 12665832 14555960, 32487457). Which type of non-hepatocytes fuse with hepatocytes in this study?

We thank the reviewer for addressing these relevant points. As we mention in the results of PHx, we did not detect any hepatocyte positive for CAG-EGFP after analysis of 11 liver lobule areas from 2 mice (4 photos per mouse) in a field of view (10x). In **Figure S5A** it is observed that CAG-EGFP staining is only expressed by non-hepatocyte cells at 0hr. However, 48 h after PHx we observe hepatocytes positive for both CAG-EGFP and Rainbow colors (mCherry, mOrange or mCerulean), as we detail for the reviewer below. Again, we do not observe any of these eGFP+, Rainbow+ cells at 0hr.

FigureS5A-PHx 0 h

48 h after PHx

48 h after PHx

48 h after PHx

As we already mentioned in the manuscript, we used a concentration of 20 mg/mL (3 consecutive days, 100 μ l per day) that resulted in hepatocyte-specific recombination throughout the liver lobule where efficiency reached 100%, with no hepatocytes positive for CAG-EGFP. We determined this TMX dose after evaluation of labelling efficiency of both cell hepatocytes and non-hepatocytes in TMX dose-response tests (**Figure S1A**) and time-course tests (**Figures 1A** and **S1B**).

As the reviewer proposed, we have now quantified the percentage of hepatocytes formed by cell fusion during the 5 time points and modified **Figure 5J** and **Supplementary Figure 5E**. We have also updated the paragraph on the discussion regarding cell fusion, addressing which type of non-hepatocyte cells could fuse with hepatocytes in our study and we added 2 new references in our manuscript (93 and 94) facilitated by the reviewer.

4. The authors state that “Taken together, these results confirm that there is not any mechanism of transdifferentiation between biliary and hepatic epithelia as we hypothetically could have found in our multicolor lineage tracing Alb-CreERT2 Rosa26rbw mice (Figures S4A-B).” It has been reported that hepatocytes could convert into biliary-like cells in cholestatic liver injury (PMID: 32971004). The authors may miss this transdifferentiation of hepatocytes into BECs because of the limitation of technology. The authors should perform co-immunostaining for fluorescent proteins and CK19 (CK19 could be set at 647 channel).

We did IHC for CK19 (**Supplementary Figure 5F**), and we did not find CK19 staining expressed by hepatocytes; CK19 was restricted expressed by bile duct cells.

Minor comments:

1. Could the authors clarify statistical methods of Fig1F. **We now explain in greater detail how we generated the graph in the Legend of Figure 1F.**

2. Does the statistical picture in Fig5H represent the sum of Ki67+ hepatocytes at 24h, 48h and 72h? yes. The author should show the number of Ki67+ hepatocytes at 24h, 48h and 72h separately.

As suggested, we have modified Figure 5H to represent the number of Ki67+ hepatocytes (PP, Mid and PC) at 24h, 48h and 72h separately.

REVIEWERS' COMMENTS

Reviewer #1 (Remarks to the Author):

Authors have adequately addressed my questions and concerns.

Reviewer #2 (Remarks to the Author):

The authors addressed my questions. The additional scRNA-seq data is useful.